# An analysis of the impact of administrative approval reform on the technological complexity of manufacturing exports

**Tingting Jiang[1], Zhibin Zhang [2]***

1 School of Economics, Shandong Technology and Business University, Yantai, Shandong, China,
2 School of Economics, Shandong Technology and Business University, Yantai, Shandong, China

* zhangzhibin2022@126.com

**Data availability statement:** The data and code for this study can be found at https://doi.org/10.6084/m9.figshare.28443707.v1.

**Funding:** This work was supported by Social Science Planning Project to Shandong Province, Research Grant number 24DJJJ16, and Shandong Technology and Business University Talent Introduction Initiation Project, Research Grant number BS202316.The funders

## Abstract

As the global economic landscape evolves, the low technological content and persistent lack of international competitiveness in China's manufacturing exports have become increasingly apparent, underscoring the urgent need for a transition from "quantity" to "quality" in the sector. Administrative approval reform, a key pillar of institutional innovation in the new era, plays a critical role in enhancing the technological complexity of manufacturing exports and strengthening international competitiveness. Using data from 2001 to 2013, this study investigates the impact of administrative approval reform on the technological complexity of manufacturing exports and explores its underlying mechanisms from both theoretical and empirical perspectives. Theoretical model analysis suggests that administrative approval reform effectively increases technological complexity by reducing the marginal and fixed costs associated with adjusting product complexity. Empirical findings provide robust evidence that administrative approval reform significantly enhances technological complexity, with results holding across various sensitivity tests. At the micro level, the reduction of institutional transaction costs emerges as a key channel through which the reform exerts its impact. Additionally, increasing investment in research and development, fixed assets, and technological innovation are identified as critical pathways influencing technological complexity. The reform's effects are particularly pronounced for non-state-owned enterprises and firms located in coastal and port cities, as revealed by a heterogeneity analysis. Furthermore, a decomposition of city-level export technological complexity using the DOP method shows that improved inter-firm resource allocation—by shifting market share from firms with lower technological complexity to those with higher technological complexity—serves as the primary mechanism driving the observed improvements at the city level. This study contributes to the literature by providing empirical evidence on the role of administrative approval reform in fostering the technological upgrading of manufacturing exports, highlighting its differentiated impact across firm types and regions. The findings offer valuable insights for policymakers seeking to enhance the technological complexity and international competitiveness of manufacturing exports in China.

had no role in study design, data collection and analysis, decision to publish, or preparation of the manuscript.

**Competing interests:** The authors have declared that no competing interests exist.

## 1. Introduction

The "Outline of the 14th Five-Year Plan for National Economic and Social Development and Vision 2035" sets forth clear development objectives, prioritizing the growth of the real economy and positioning China as a leading global manufacturing powerhouse. A key component of this strategy involves enhancing the technological complexity of manufacturing exports, which is crucial not only for achieving these goals but also for addressing challenges arising from diminishing labor cost advantages and growing international trade frictions. However, excessive institutional transaction costs have emerged as a significant barrier to improving the technological complexity of China's manufacturing exports, and the problems in the import and export commodity inspection process are particularly prominent.

The inspection of import and export commodities refers to the inspection and identification of indicators such as the quality, specifications, weight, quantity, packaging, safety performance, and sanitary conditions of import and export commodities, as well as the inspection of transportation technology and conditions, in order to determine whether they are consistent with trade contracts and relevant standard provisions, and whether they comply with the relevant laws and administrative regulations of the importing and exporting countries. According to the "Catalog of Commodities Subject to Inspection, Quarantine, and Testing by Entry-Exit Inspection and Quarantine Organizations," inspection departments are required to inspect commodities based on specified regulations. Only those that meet the criteria of the "Four Laws and Three Regulations" are eligible to clear customs. The "Four Laws and Three Regulations" include the following: the Law of the People's Republic of China on the Inspection of Import and Export Commodities, the Law on the Quarantine of Animals and Plants upon Entry and Exit, the Law on Frontier Health Quarantine, and the Law on Food Safety (the "Four Laws"); as well as the Regulations on the Implementation of the Law on the Inspection of Import and Export Commodities, the Regulations on the Implementation of the Law on the Quarantine of Animals and Plants, and the Detailed Rules for the Implementation of the Law on Frontier Health Quarantine (the "Three Regulations").

The primary aim of these inspections, as stipulated by the relevant laws, is to protect public interests, including food safety and biosecurity, rather than focusing exclusively on product quality. However, in practice, widespread large-scale quality inspections for imported and exported commodities often result in overlapping government functions, bureaucratic disputes, and avoidance of responsibility. This situation significantly hinders the efficiency of goods circulation and increases business costs. Additionally, inconsistent testing standards grant considerable discretionary power to inspection departments, creating opportunities for rent-seeking behaviors and even corruption.

In summary, manufacturing enterprises face substantial institutional transaction costs during the import and export process, which increase the burden on foreign trade enterprises and hinder the enhancement of technological complexity in manufacturing exports.

According to "From import and export commodity inspection to deepening approval reform", the annual inspection costs for import and export commodities nationwide exceed 10 billion yuan, while related additional charges exceed 30 billion yuan, accounting for approximately 5% of the profits of foreign trade enterprises.

In this context, administrative approval reform has become a crucial factor in reducing institutional transaction costs, unlocking institutional dividends through structural and procedural changes, and enhancing the technological complexity of manufacturing exports. Since the establishment of my country's first administrative approval center in Shenzhen in 1995, the number of administrative approval centers has reached 316 by 2015, playing an important role in transforming government functions, reducing institutional transaction

costs, and stimulating market vitality (Fig 1). Correspondingly, Correspondingly, the technological complexity of my country's manufacturing exports continued to increase from 2001 to 2013, and the two showed an obvious positive correlation (Fig 2). It is evident that administrative approval reform can effectively promote the technological complexity of manufacturing exports by reducing institutional transaction costs. However, the existing literature remains unclear on the precise relationship between administrative approval reform and the technological complexity of manufacturing exports, with few studies systematically examining its impact and underlying mechanisms. Therefore, this study not only offers insights into enhancing the technological complexity of manufacturing exports through the lens of

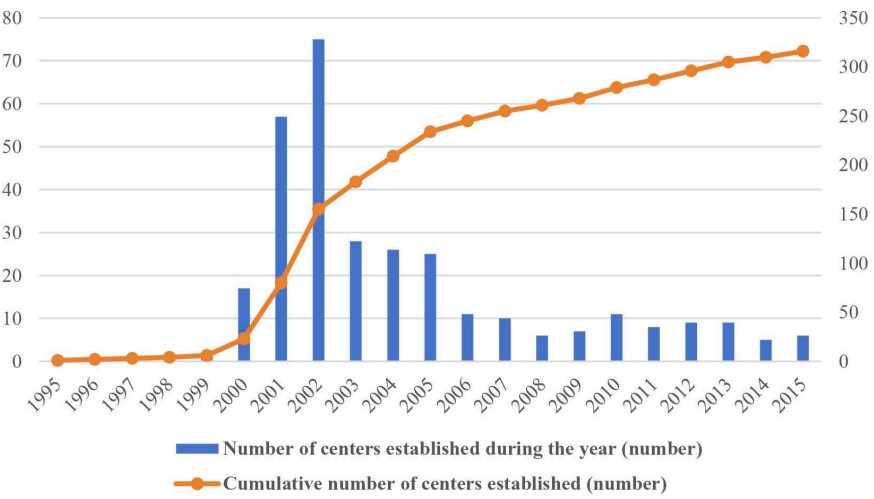

**Fig 1. Establishment of Administrative Approval Centers from 1995 to 2015.** This figure provides a clear visual representation of the gradual development of these centers over the 20-year period, from non-existence to establishment, offering an important timeline and context for analyzing the impact of related policies.

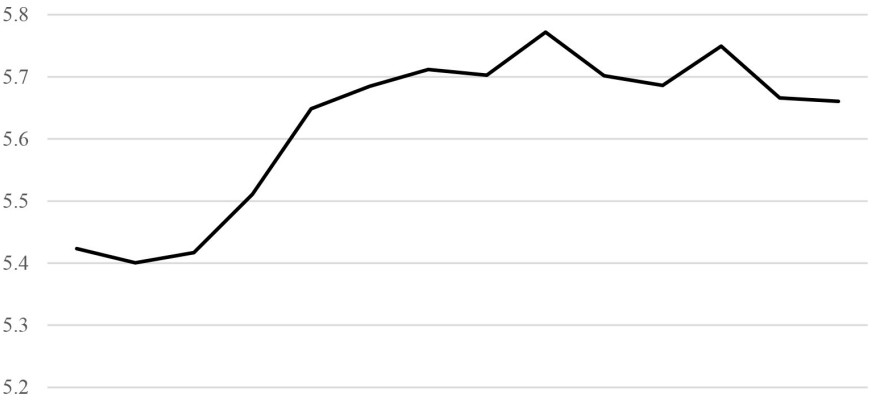

**Fig 2. Technological complexity of manufacturing exports, 2001-2013.** This figure visualizes the data on technological complexity across different years, allowing us to clearly observe the changing trends in the technological level of manufacturing exports during this period. It is of significant importance for assessing the technological competitiveness of the manufacturing sector in the international market.

institutional transaction costs but also holds significant practical implications for advancing reforms in "streamlining government functions," optimizing governmental roles, and fostering high-quality manufacturing development.

## 2. Literature review

The administrative approval system reform, a critical mechanism for reshaping the business environment, has garnered significant attention from both the economic research community and policymakers in recent years. Zhu and Zhang (2016) were pioneers in exploring the diffusion process of China's administrative approval system reform, analyzing the determinants influencing local governments' decisions to establish administrative approval centers [1]. Building on this foundation, subsequent research has delved into the economic benefits derived from the reform from various perspectives. Xia and Liu (2017), through robust empirical analysis, identified a significant positive correlation between administrative approval system reform and overall economic growth [2]. By streamlining approval procedures and reducing administrative interference, the reform has effectively increased the participation of market entities and improved resource allocation efficiency, injecting new momentum into macroeconomic growth. Liu and Zhong (2018) further highlighted the pivotal role of administrative approval reform in addressing overcapacity issues [3]. Under the traditional administrative approval framework, local governments, driven by factors such as promotion incentives, often overexercised their approval powers to facilitate capacity expansion. The reform, by curbing the improper exercise of such powers, has guided capacity structures toward a more rational, efficient, and sustainable optimization pathway, thereby promoting the structural adjustment and upgrading of the broader economy. Lai et al. (2018) underscored the critical role of administrative approval system reform in optimizing market resource allocation [4]. The reform reduces the dispersion in productivity distribution across cities, facilitates a more balanced and efficient flow of resources across regions and industries, and ultimately enhances the operational efficiency and stability of the economic system.

At the enterprise level, the administrative approval system reform has also instigated profound and transformative impacts. Wang and Feng (2018) employed the difference-in-differences model to test and found that the administrative approval system reform has remarkably promoted the elevation of enterprise innovation levels and the optimization of production efficiency [5]. After the reform, the simplification and optimization of the approval process have substantially reduced the time costs and energy expenditures of enterprises in key operational aspects such as project approval and qualification acquisition, enabling enterprises to allocate more resources to core competitiveness-building areas like R & D innovation and production process improvement, thereby vigorously enhancing the core competitiveness of enterprises in market competition. Chen (2015) found that the decline in regional administrative approval intensity can significantly lower the cost threshold for enterprises to initiate entrepreneurial activities and reduce the institutional barriers to enterprise access [6]. This transformation has created a more relaxed and favorable institutional environment for the birth and growth of emerging enterprises, significantly increasing the probability of enterprises entering the market and effectively promoting the formation of a diversified pattern of market entities and the intensification of market competition. Zhang et al. (2016) further discovered that the administrative approval system reform not only affects the enterprise establishment stage but also can remarkably augment the individual entrepreneurship probability and expand the local entrepreneurship scale, comprehensively stimulating the entrepreneurial vitality and innovation spirit of the whole society and cultivating more new growth points for macroeconomic development [7]. Additionally, Cheng et al. (2023) demonstrated that the optimization of the administrative

approval process can directly stimulate a significant increase in enterprise investment levels [8]. The administrative approval system reform, by enhancing government administrative efficiency, effectively reduces the information cost obstacles faced by enterprises in the investment decision-making process, augments the flexibility and precision of investment decisions, and renders enterprises more proactive and scientific in resource allocation decision-making. Zheng et al. (2023) and Zhu et al.(2021) further corroborated that the administrative approval reform has a significant promoting effect on the improvement of enterprise productivity. Although certain differences may exist among different industries, regions, and enterprise ownership types, in general, the reform has laid a solid foundation for the enhancement of enterprise comprehensive competitiveness through multiple channels such as optimizing resource allocation efficiency, promoting the diffusion of technological innovation, and improving enterprise internal management efficiency [9,10].

In the field of international trade research, the significance of the institutional environment and business environment has also been widely and deeply recognized. In 1990, North pioneered the classic proposition of "institutions initiate trade", highlighting the fundamental and leading role of institutions in the international trade system. Since then, numerous scholars have further delved into the theoretical connotations and empirical evidences of the specific impact mechanisms of institutional quality on international trade. It has been found that institutional comparative advantage has become one of the key factors determining the choice of international trade modes and the pattern of division of labor. Differences in institutional arrangements among different countries or regions, such as the degree of perfection of intellectual property protection systems and the stability and predictability of trade policies, profoundly influence their strategic choices of specialized division of labor positioning and trade mode in international trade (Yang, 2019 [11]). Dai and Jin (2014) [12], from the perspective of division of labor evolution, found that institutional quality has a significant promoting effect on the technological complexity of a country's export products. Meanwhile, the degree of intra-product international division of labor and its interaction with institutional quality also have a significant promoting effect on the enhancement of export technological complexity. Dai and Zheng (2015) further revealed that there is a close internal relationship between institutional quality and the division of labor position of a country's manufacturing industry in the global value chain [13]. A high-quality institutional environment can effectively attract more high-end production factors, drive the manufacturing industry to ascend and transform towards high value-added links, and thus occupy a favorable strategic position and competitive advantage in the global value chain competition (Hu and Zhang, 2015; Du and Peng, 2019 [14,15]). Additionally, Zhang and Yang (2022) found that the optimization of the business environment can improve the position of private enterprises in the value chain [16]. Dai (2020) believed that this positive promoting effect stems not only from the direct effect of business environment optimization but also from the indirect effect formed through intermediary roles such as value chain migration, innovation activity enhancement, and trade condition improvement [17].

Specifically regarding the administrative approval system reform, Nunn (2007) was the first to propose that the administrative approval efficiency of a country (region) has a significant impact on domestic economic growth and foreign trade comparative advantage [18]. The level of administrative approval efficiency is directly related to the enterprise's operating cost structure, innovation speed, and market response ability, and thus is manifested as differential performance of comparative advantage in the international trade competition pattern (Feng et al., 2018; Liu and Shen, 2019; Mao and Hu, 2024 [19–21]). Feng et al. (2018) empirically found that the establishment of administrative approval centers can effectively promote more manufacturing enterprises to enter the international market to participate in competition [19].

Although the effect of policy reform may have a time lag, from a long-term dynamic perspective, it can significantly expand the export scale and market share of cities and industries. On this basis, Liu and Shen (2019) further conclusively confirmed that after China's administrative approval system reform, the division of labor position of manufacturing enterprises in the global value chain has shown a significant upward trend [20]. Mao and Hu (2024) also found that the administrative approval reform has increased the general trade export ratio of enterprises and has a significant promoting effect on the transformation and upgrading of export trade [21]. Alleviating financing constraints, intensifying market competition, and improving enterprise innovation ability are important channels through which the administrative approval reform promotes the transformation and upgrading of export trade. Xu and Wang (2018) took the decentralization of export tax rebate (exemption) approval power in 2006, a specific administrative approval system reform in China, as a quasi-natural experiment and found that the experimental group enterprises have achieved significant improvements in key indicators such as export scale, product quality, and export performance [22].

Overall, existing research on the relationship between administrative approval reform and the technological complexity of manufacturing exports remains insufficient, particularly in terms of direct analysis and quantitative assessment of how such reforms contribute to enhancing export technological complexity. Although previous studies have provided partial insights into the link between institutional quality and the technological complexity of manufacturing exports, offering valuable theoretical perspectives on the impact of administrative approval reform, the specific mechanisms underlying this relationship require further comprehensive and in-depth investigation.

Compared with existing research, this study makes the following three key contributions: (1) Innovative Research Perspective: This study empirically examines the impact of administrative approval reform on the technological complexity of manufacturing exports using enterprise-level microdata. By addressing the limitations of prior research, it provides a more nuanced understanding of the relationship between administrative reforms and export technological complexity. (2) Theoretical Contribution: Extending the model proposed by Hallak and Sivadasan (2013), this study conceptualizes administrative approval reform as a mechanism for reducing institutional transaction costs [23]. By lowering the marginal and fixed costs associated with adjusting product complexity, the reform influences firms' decisions on technological complexity. This approach elucidates the underlying mechanisms through which administrative approval reform impacts the technological sophistication of exports. (3) Empirical Insights: The study confirms that administrative approval reform significantly enhances the technological complexity of manufacturing exports. Moreover, it explores the heterogeneous effects of these reforms across cities with varying characteristics, offering a deeper understanding of regional disparities. Additionally, it investigates the micro-level mechanisms underlying the relationship, focusing on the reduction of institutional transaction costs, innovation stimulation, and resource reallocation efficiency.

## 3. Theoretical analysis

We have examined three cases of enterprises benefiting from Guangxi's "full-chain approval service" for entrepreneurship. Case 1: Guangxi Changhong Pharmaceutical Co., Ltd. reduced its required submission materials from 26 items to 23, shortened the processing time from 35 working days to 10 working days, and reduced the number of in-person visits from 3 to 1. Case 2 and Case 3: Both Guangxi Chunjiang Food Co., Ltd. and Guangxi Huxin Import & Export Co., Ltd. experienced a reduction in submission materials from 14 items to 12, a decrease in processing time from 35 to 10 working days, and a reduction in in-person visits

from 2 to 1. Overall, these enterprises saw significant streamlining of their approval processes, with an average reduction of 25 working days in processing time, along with a notable decrease in submission requirements and in-person visits. This led to a substantial increase in administrative efficiency, alleviating the burden on enterprises and facilitating their rapid growth. Additionally, Liang et al. (2020) [24], through a questionnaire survey on the paperless reform of the export tax rebate policy, further corroborated the positive impact of administrative approval reforms on manufacturing enterprises.

Drawing on existing literature and the aforementioned case analyses, reducing institutional transaction costs and effectively leveraging the resource allocation effects of market mechanisms are identified as the primary channels through which administrative approval reform enhances economic performance [19,20]. Therefore, this paper conceptualizes administrative approval reform as a policy instrument for reducing institutional transaction costs and incorporates it into Hallak and Sivadasan's (2013) theoretical model to examine its impact and underlying mechanisms on the technological complexity of manufacturing exports [23].

(i) Household sector

Assuming that the utility function of the representative consumer takes the CES form:

$$u = \left( \int_{g \in \Theta} \left( \lambda_g q_g \right)^{\frac{\sigma-1}{\sigma}} dg \right)^{\frac{\sigma}{\sigma-1}}$$

(1)

$$s.t \quad E = \sum_g p_g q_g$$

(2)

where g denotes the type of consumer product; $\Theta$ is the set of product types; $\sigma (\sigma > 1)$ is the elasticity of substitution between different consumer products; and $\lambda$ and $q$ denote the technological complexity and quantity of products for consumer product type g, respectively. Equation (2) represents the budget constraint faced by a representative consumer. The expression of the representative consumer's demand function for product category g can be obtained by solving for consumer utility maximization:

$$q_g = p_g^{-\sigma} \lambda_g^{\sigma-1} \frac{E}{P}$$

(3)

where $P$ denotes the total price index, i.e., $P = \left( \int_{g \in \Theta} p_g^{1-\sigma} \lambda_g^{\sigma-1} dg \right)^{\frac{1}{1-\sigma}}$. In order to make the theoretical formula more concise, the symbol for product g is omitted below.

(ii) Production sector

We assume that each enterprise produces only one type of consumer product. Building on the theoretical model of Hallak and Sivadasan (2013) [23], we assume heterogeneity among enterprises in terms of both productivity and production capacity. Higher productivity leads to lower marginal costs, while stronger production capacity, given the technological complexity of the product, results in lower fixed costs. Based on the above assumptions, the expression for an enterprise's fixed costs is given by:

$$FC(\lambda, \xi) = F_0 + \frac{f}{\xi} \lambda^{\beta}$$

(4)

Where, $F_0$ is a constant, indicating the unconditional fixed cost required for the production of the enterprise, $f$ is a constant, $\beta$ is the technological complexity elasticity coefficient of fixed

cost, indicating the impact of the enterprise's upgrading of the technological complexity of the product on the fixed cost; and $\xi$ indicates the production capacity.

The expression for the marginal cost of a firm is:

$$MC(\lambda,\ \varphi)=\frac{c}{\varphi}\lambda^{\alpha} \tag{5}$$

where $\varphi$ is the firm productivity, $\alpha$ is the technological complexity elasticity of marginal cost, which indicates the effect of upgrading the technological complexity of the product on the marginal cost, and $c$ is a constant.

Based on the previous analysis, the administrative approval reform can not only save the approval time by optimizing the approval process and realizing multi-departmental collaboration, but also standardize the approvals including commodity inspection catalog and inspection charges, reduce the systematic transaction costs, and help enterprises to spend more time and capital on directly improving the technological complexity of products $\lambda$ and their production capacity $\xi$. Assuming that $\theta$ is the intensity of administrative approval reform, the value range of $\theta$ is $[0,\ \bar{\theta}\ ]$, $\bar{\theta}$ indicates the maximum value of the intensity of the implementation of administrative approval reform, the larger the value of $\theta$, the higher the intensity of the administrative approval reform, and when $\theta=\bar{\theta}$, it means that the systematic transaction cost is zero. Based on this, after the implementation of administrative approval reform, the product technological complexity and production capacity expressions are $\bar{\lambda}=(1-\theta)\lambda$ and $\bar{\xi}=(1+\theta)\xi$ respectively, which means that the higher the intensity of administrative approval reform, the lower the marginal cost and fixed cost of enterprises to improve the technological complexity of the products, and the fixed cost and marginal cost of enterprises after the implementation of administrative approval reform can be obtained by bringing them into Eq. (4) and Eq. (5):

$$FC=F_{0}+\frac{f}{\xi(1+\theta)}(1-\theta)^{\beta}\lambda^{\beta} \tag{6}$$

$$MC=\frac{c}{\varphi}(1-\theta)^{\alpha}\lambda^{\alpha} \tag{7}$$

(iii) Market equilibrium

We assume a monopolistically competitive market structure in the production sector, where firms employ cost-plus pricing:

$$p=\frac{\sigma}{\sigma-1}\frac{c}{\varphi}\big[(1-\theta)\lambda\big]^{\alpha} \tag{8}$$

Furthermore, an expression for the firm's profit can be derived as follows:

$$\pi=q(p-MC)-F \tag{9}$$

After bringing Eqs. (3), (6), (7), and (8) into Eq. (9), the collation can be obtained:

$$\pi=\frac{\sigma^{-\sigma}}{(\sigma-1)^{1-\sigma}}\frac{E}{P^{1-\sigma}}\left(\frac{c}{\varphi}\right)^{1-\sigma}\big[(1-\theta)\lambda\big]^{\beta+\sigma-1}-F_{0}-\frac{f}{\xi(1+\theta)}\big[(1-\theta)\lambda\big]^{\beta} \tag{10}$$

Derivation of product technological complexity $\lambda$ on both sides of equation (9) yields an expression for the optimal product technological complexity in market equilibrium:

$$\lambda = \left[ \frac{1-\alpha}{\beta} \left( \frac{\sigma-1}{\sigma} \right)^{\sigma} \left( \frac{\varphi}{c} \right)^{\sigma-1} \frac{(1-\theta)\xi}{f} \frac{E}{P} \right]^{\frac{1}{t}} \frac{1}{1-\theta} \tag{11}$$

Where, $t = \beta - (1-\alpha)(\sigma-1)$, according to the theoretical model of Hallak and Sivadasan (2013) [23], it is assumed that $t \in (0,1)$. From equation (11), the technological complexity of the product in the market equilibrium depends on the change of $\theta$, which is obtained by the derivation of both sides of equation (10):

$$\frac{\partial \lambda}{\partial \theta} = R\left[ (1-t) - (1+t)\theta \right] \tag{12}$$

Where, $R = \left[ \frac{1-\alpha}{\beta} \left( \frac{\sigma-1}{\sigma} \right)^{\alpha} \left( \frac{\varphi}{c} \right)^{\sigma-1} \frac{\xi}{f} \frac{E}{P} \right]^{\frac{1}{t}} \frac{(1+\theta)^{\frac{1}{t}-1}}{t(1-\theta)^2} > 0$, and thus, the sign of $\frac{\partial \lambda}{\partial \theta}$ mainly

depends on $\left[ (1-t) - (1+t)\theta \right]$. Further based on the relationship between $\bar{\theta}$ and $\frac{1-t}{1+t}$, when

$\bar{\theta} < \frac{1-t}{1+t}$, the following research conclusion can be obtained, as the intensity of the admin-

istrative approval reform increases, the technological complexity of the product is increas-

ing; and when $\bar{\theta} > \frac{1-t}{1+t}$, the following research conclusion can be obtained, when $\theta \in [0,$

$\frac{1-t}{1+t}$ ]case, as the intensity of the administrative approval reform increases, the technological

complexity of the product will increase; and when [$\theta \in \frac{1-t}{1+t}$ , $\bar{\theta}$ ]case, as the intensity of the

administrative approval reform increases, the technological complexity of the product will decrease. complexity will decrease.

Upon re-examination of Fig 1 and Fig 2, it becomes evident that the theoretical model

corresponding to the range $\theta \in [0, \frac{1-t}{1+t}$ ] can explain the positive correlation between

administrative approval reform and product technological complexity observed from 2001 to 2013. This indicates that administrative approval reform effectively promotes the enhancement of manufacturing export technological complexity by reducing the marginal and fixed costs associated with adjusting product technological complexity. It is important to note that the theoretical model's conclusions also suggest that the positive impact of administrative approval reform on enhancing manufacturing export technological complexity may not persist indefinitely. As the marginal cost of enterprises in enhancing product technological complexity falls too low, competition surrounding product complexity intensifies, ultimately leading to declining prices and profits. At this stage, further intensification of administrative approval reform may result in a reduction in product technological complexity. Next, we will employ econometric methods to empirically test the impact and underlying mechanisms of administrative approval reform on the technological complexity of manufacturing exports.

## 4. Method

### 4.1. Empirical model

(1) The Baseline Regression Model

To empirically examine the impact mechanism of administrative approval reform on the technological complexity of manufacturing exports, we develop the following econometric model:

$$lnY_{ijt} = \beta_0 + \beta_1 ALC_{jt} + \beta Z_{ijt} + v_i + v_j + \mu_{ijt} \tag{13}$$

Here, $i$, $j$, and $t$ denote the enterprise, region, and year, respectively. $Y$ denotes the technological complexity of manufacturing exports, $ALC$ indicates the variable for administrative approval reform, and $Z$ captures a series of control variables that may influence the technological complexity of manufacturing exports, including variables at the enterprise, and city levels. Additionally, we incorporate enterprise fixed effects $v_i$ and time fixed effects $v_j$ Into the model to control for factors that remain constant across entities and over time, which could otherwise confound the estimation results. $\mu_{ijt}$ denotes the random error term.

(2) Mechanism Analysis Model

To further explore the impact of administrative approval reform on the technological complexity of manufacturing exports, this study employs regression equations to identify the specific mechanisms through which the reform influences export technological complexity.

$$lnM_{ijt} = \beta_0 + \beta_1 ALC_{jt} + \beta Z_{ijt} + v_i + v_j + \mu_{ijt} \tag{14}$$

The variable $M$ represents institutional transaction costs, R&D investment, fixed asset investment, and technological innovation. By conducting a regression on Equation (14), we can assess, based on the resulting coefficients, whether administrative approval reform impacts the technological complexity of firms' exports through institutional transaction costs, R&D investment, fixed asset investment, and technological innovation.

(3) Parallel trend test

Building on Equation (13), we introduce dummy variables to capture the pre-reform and post-reform periods of administrative approval, resulting in the following econometric model:

$$lnY_{ijt} = \beta_0 + \sum_{m=1}^{2}\beta_m ALC\_pre_{c,t-m} + \sum_{n=0}^{4}\beta_n ALC\_post_{c,t+n} + \beta Z_{ijt} + v_i + v_j + \mu_{ijt} \tag{15}$$

In the equation, $ALC\_pre$ represents the pre-reform period of administrative approval. Here, $m=1$ corresponds to the year immediately preceding the administrative approval reform, and $m=2$ represents the second year preceding the reform. $ALC\_post$ signifies the post-reform period of administrative approval, where $n=1$ corresponds to the first year of the reform, $n=2$ corresponds to the second year, and so forth. The regression coefficient of $ALC\_pre$ in Equation (16) is our focal point of interest. If the coefficient of $ALC\_pre$ is close to zero, it indicates that before the establishment of administrative approval centers, there is a similar trend in the change of technological complexity in manufacturing exports between cities with and without administrative approval centers, suggesting no significant difference between them before the reform.

## 4.2 Indicator selection

(1) Dependent variable

The technological complexity of manufacturing exports is a focal point of our investigation. Drawing on the approaches used in literature such as Hausmann et al. (2007) [25], Yue et al. (2016) [26], Lu and Jin(2020) [27], we calculated the technological complexity of manufacturing exports for enterprises. The measurement process is as follows:

Step 1 involves calculating the technological complexity of a specific product based on Equation (16):

$$\mathrm{Pr}ody_k = \sum_j \frac{\left(x_{kj} / X_j\right)}{\sum_j \left(x_{kj} / X_j\right)} G_j \tag{16}$$

In this equation, $k$ denotes the product category, $j$ signifies the country of origin, $x_{kj}$ represents the transaction amount for product $k$ exported by country $j$, $X_j$ represents the overall export value of country $j$, and $G_j$ stands for the per capita GDP of country $j$.

Step 2 involves calculating the technological complexity of manufacturing exports for enterprises based on Equation (17) after obtaining the technological complexity for product $k$:

$$Y_i = \sum_k \frac{x_{ik} \mathrm{Pr}ody_k}{X_j} \tag{17}$$

(2) Independent variable

We consider administrative approval reform a key policy measure aimed at enhancing administrative efficiency. The establishment and expansion of administrative approval centers represent a concerted effort in China's administrative approval reform [19]. By consolidating approval tasks previously dispersed across various departments, this reform reduces the commuting costs for enterprises, effectively shortening the approval cycle and lowering institutional transaction costs. Therefore, in line with existing literature, we create a dummy variable based on the timing of administrative approval center establishment. If city $j$ establishes an administrative approval center in year $t$, the variable is assigned a value of 1; otherwise, it is assigned a value of 0.

When assessing administrative approval reform solely from a temporal perspective, there is a risk of overlooking the heterogeneous impacts arising from variations in reform intensity. In practice, even when different regions implement administrative approval reform simultaneously, substantial differences in reform intensity can occur due to disparities in the number of departments, approval items, and service windows within administrative approval centers. For example, in 2001, both Zhoushan City and Lishui City implemented administrative approval reforms. The administrative approval center in Zhoushan City comprised 26 departments (including both approval and service departments), 685 approval items (including both approval and service items), and 79 service windows. In contrast, the administrative approval center in Lishui City consisted of only 27 departments, with no approval items or service windows. This stark contrast clearly demonstrates that the intensity of administrative approval reform in Zhoushan City was significantly higher than in Lishui City.

As a general principle, a greater number of departments or approval items involved indicates a higher intensity of administrative approval reform, which in turn leads to a more significant reduction in institutional transaction costs. Therefore, we use the number of

departments in the administrative approval center (encompassing both approval and service departments) and the number of approval items (including both approval and service items) as alternative indicators for robustness checks.

(3) Mediating Variables

Theoretical analysis identifies institutional transaction costs, per capita institutional transaction costs, R&D investment, fixed asset investment, and technological innovation as key mechanisms through which administrative approval reform impacts the technological complexity of manufacturing exports.

Institutional Transaction Costs (ITC) and Per Capita Institutional Transaction Costs (PITC): The reduction in institutional transaction costs is the most direct outcome of administrative approval reform. To measure these costs, this study adopts the approach outlined by Guo and Shao (2019) [28], which calculates the sum of management, financial, and sales expenses, along with their average per employee, to assess both ITC and PITC.

Fixed Asset Investment (FAI): Innovation activities and enhancements in operational efficiency rely on continuous R&D efforts and sustained fixed asset investment. Fixed asset investment is calculated as the difference between total fixed assets in the current year and the previous year, plus current-year depreciation.

R&D Investment (R&D): Measured as the logarithm of a company's R&D expenditure. Generally, higher R&D investments foster innovation, which subsequently enhances the technological complexity of a company's exports.

Technological Innovation (INV): Informed by previous research, technological innovation is measured as the logarithm of the output value of new products. A higher value indicates greater innovation efforts, which positively influence the technological complexity of exported products.

(4) Control variables

In this study, we selected the following control variables to ensure a more comprehensive and accurate analysis of the impact of administrative approval system reform on the technological complexity of manufacturing exports:

Firm age (Age): Reflects the number of years since the company was established and is measured by the logarithm of the company's age. Older companies may have accumulated more experience and resources, which can influence their capacity for technological innovation and the complexity of their exported products.

Industry structure (Sec): Represented by the proportions of secondary, and tertiary industries. The technological requirements vary significantly across these sectors, and optimizing the industry structure, particularly by increasing the share of high-tech industries, can enhance the technological complexity of exported products.

Firm size (Scale): Measured by the logarithm of highway freight volume, improved transportation conditions help companies access international markets more quickly, thereby enhancing the technological sophistication of their exported products.

Transportation convenience (Commdity): Assessed by highway freight volume. Improved transportation conditions facilitate faster access to international markets and raise the technological sophistication of exported goods.

Debt-to-asset ratio (Debt): A key indicator of financial health. Excessive debt may limit a company's ability to invest in technology, thus impacting the complexity of its exported products.

Wage Level (Sarsy): Influences labor costs and the company's ability to attract high-quality talent. Higher wage levels may enable firms to recruit more innovative personnel, thereby enhancing technological complexity.

Population density (Peo): Measured by the ratio of urban population to urban area. Higher population density often indicates greater availability of resources and larger market opportunities, which can stimulate technological innovation and complexity.

These control variables provide a more detailed perspective for analyzing the impact of administrative approval system reforms on the technological complexity of manufacturing exports, helping to reveal the true effects of these reforms.

## 4.3. Sample selection and data

The relevant indicators are sourced from the China Industrial Enterprise Database, the China Customs Database, the BACI Bilateral Trade Statistics Database, the World Bank Database, and the China Economic Network Statistics Database. Specifically, the China Industrial Enterprise Database, the China Customs Database, the BACI Bilateral Trade Statistics Database, and the World Bank Database are primarily used to calculate the technological complexity of manufacturing exports. The China Economic Network Statistics Database is utilized to control for city-level factors that may influence the technological complexity of manufacturing exports.

The data on administrative approval reform are sourced from the county-level administrative approval center database developed by Professor Xu Xianxiang's research team at Lingnan College, Sun Yat-sen University. This dataset covers the period from 1995 to 2015. Administrative approval centers were established nearly every year after 2001 (Fig 1). Due to limitations in the data from the "China Industrial Enterprise Database," this study selects the sample period from 2001 to 2013 to examine the impact of administrative approval system reform on the technological complexity of manufacturing exports.

The data processing and matching process for the technological complexity of manufacturing exports involve three main steps:

Step 1: Calculate the Dependent Variable, Export Technological Complexity. To calculate the technological complexity of products, as specified in formula (14), global per capita GDP data and bilateral trade data from various countries are required. The BACI Bilateral Trade Statistics Database compiles bilateral trade data and values for HS6-coded products from 246 countries and regions worldwide. This dataset is then matched with the per capita GDP data of these countries and regions, sourced from the World Bank database, to compute the technological complexity of HS6-coded products. Subsequently, the technological complexity of HS6-coded products is matched with data from the China Customs database, and the manufacturing export technological complexity at the enterprise level is calculated according to formula (15).

Step 2: Match Customs Data with Industrial Enterprise Data. Before matching the customs data with the industrial enterprise database, we exclude samples that lack enterprise names, destination country names, or product names. Additionally, we remove samples with a single transaction size below $50, or a quantity less than 1, as well as those involving agricultural or resource products and trade intermediaries. Following the methodology of Su et al. (2018) [29], we first match the data based on enterprise names and years. For the remaining samples, we proceed with matching using the last seven digits of the phone number and the postal code.

Step 3: Processing the Industrial Enterprise Database. Following the methodologies outlined by Nie et al. (2012) and Brandt et al. (2012) [30,31], we processed and filtered the samples from the industrial enterprise database. Specifically, we undertook the following steps:

(1) Outlier Removal: We removed outliers and missing values related to industrial value-added, sales revenue, and the number of employees. This included eliminating negative values, instances where fixed capital exceeded current capital, and enterprises that did not meet the criteria for large-scale enterprises.

(2) Exclusion of Specific Industries: We excluded enterprises classified under the "mining" and "electricity, gas, and water production and supply" industries, as identified by their 2-digit industry codes.

(3) Geographical Exclusion: We also excluded industrial enterprises located in the four municipalities of Beijing, Shanghai, Chongqing, and Tianjin.

Table 1 reports the descriptive statistics of the main variables.

## 5. Empirical result analysis

### 5.1. Benchmark results

To mitigate the potential impact of the sample distribution of the dependent variable on the test results, we excluded the top and bottom 1% of the sample data. In addition, to address potential issues related to serial correlation and heteroskedasticity, we used robust standard error estimates, clustered at the enterprise level. Table 2 presents the baseline regression results. In column (1), only the core explanatory variable is included. In columns (2) through (4), we control for both enterprise and time fixed effects while progressively introducing control variables at the enterprise level. Across all columns in Table 2, it is evident that the estimated coefficients of alc_dum are significantly positive at the 1% significance level. This finding indicates that administrative approval reform has a significant positive impact on the technological complexity of manufacturing exports. For example, in column (4), the establishment of administrative approval centers is associated with a 0.33% increase in the technological complexity of manufacturing exports.

### 5.2. Robust test

(1) Replace independent variables

Solely considering the time dimension may not provide a comprehensive measure of the impact of administrative approval reform on the technological complexity of manufacturing exports. To address this, we re-examined the technological complexity of manufacturing exports by incorporating the number of administrative approval center departments

**Table 1. Descriptive statistics.**

| Variable | Variable Description | Obs | Mean | Std.Dev. | Min | Max |
|---|---|---|---|---|---|---|
| lfirmprody | Technological Complexity | 668000 | 5.705 | 0.72 | 1.469 | 7.568 |
| alc_dum | Administrative Approval Reform | 682000 | 0.71 | 0.454 | 0 | 1 |
| age | Firm Age | 682000 | 2.12 | .658 | 0 | 7.607 |
| scale | Firm Size | 679000 | 5.378 | 1.125 | 2.079 | 12.316 |
| debt | Debt-to-Asset Ratio | 606000 | 0.45 | 0.311 | 0 | 1 |
| sec | Industrial Structure | 682000 | 51.172 | 8.182 | 8.05 | 92.3 |
| peo | Population Density | 678000 | 1.285 | .683 | .196 | 9.542 |
| sarsy | Wage Level | 677000 | 14.291 | 1.448 | 9.274 | 17.976 |
| commdity | Transportation Accessibility | 682000 | 9.23 | .839 | 2.303 | 13.225 |
| ITC | Institutional transaction costs | 445000 | 8.536 | 1.466 | 0 | 18.322 |
| PITC | per capita institutional transaction costs | 443000 | 3.197 | 1.08 | 0 | 12.43 |
| R&D | Fixed asset investment | 107000 | .937 | 2.345 | 0 | 15.782 |
| FAI | R&D investment | 197000 | 7.968 | 1.955 | 0 | 17.139 |
| INV | Technological innovation | 271000 | 1.058 | 3.121 | 0 | 18.015 |

Table 2. Benchmark regression results.

| VARIABLES | (1) | (2) | (3) | (4) |
|---|---|---|---|---|
| | lfirmprody | lfirmprody | lfirmprody | lfirmprody |
| alc_dum | 0.0186*** | 0.0044*** | 0.0037*** | 0.0033*** |
| | (0.0010) | (0.0011) | (0.0011) | (0.0011) |
| age | | | -0.0211*** | -0.0217*** |
| | | | (0.0031) | (0.0031) |
| scale | | | -0.0065*** | -0.0068*** |
| | | | (0.0018) | (0.0018) |
| debt | | | 0.0029 | 0.0039 |
| | | | (0.0038) | (0.0039) |
| sec | | | | 0.0012*** |
| | | | | (0.0003) |
| peo | | | | -0.0008 |
| | | | | (0.0020) |
| sarsy | | | | -0.0016 |
| | | | | (0.0035) |
| commdity | | | | 0.0021 |
| | | | | (0.0037) |
| Constant | 5.6639*** | 5.6978*** | 5.7713*** | 5.7154*** |
| | (0.0030) | (0.0023) | (0.0116) | (0.0636) |
| Year FE | NO | YES | YES | YES |
| Firm FE | NO | YES | YES | YES |
| Observations | 668,488 | 628,079 | 552,090 | 542,551 |
| $R^2$ | 0.0022 | 0.7802 | 0.7783 | 0.7792 |

Robust standard errors clustered at the enterprise level are reported in parentheses. "Yes" indicates the inclusion of relevant variables, while "No" indicates their exclusion. *, **, *** denote significance levels at 1%, 5%, and 10%, respectively. The same notation applies below.

(alc_department) and the number of approval items (alc_item) as explanatory variables. The results are presented in Table 3. Columns (1) and (2) display the test results. The estimated coefficients of alc_department and alc_item are both significantly positive at the 1% level. This indicates that an increase in the number of departments and approval items contributes to enhancing the technological complexity of manufacturing exports. Economically, for every 1% increase in the number of departments and approval items, the technological complexity of manufacturing exports is expected to increase by 1.2% and 1%, respectively.

(2) Robustness testing for individual years

Drawing inspiration from the methodology of Wang and Feng (2018) [5], we designate the year 2002 as the policy shock year for administrative approval reform. Cities that established administrative approval centers in 2002 are categorized as the experimental group, while cities that either did not establish such centers before 2013 or did so after 2013 constitute the control group. We conducted regression analyses based on these criteria, and the results are presented in column (3) of Table 3. The estimated coefficient for the interaction term *2002treat × alc* is found to be significantly positive, at least at the 10% significance level. This indicates that the establishment of administrative approval centers in 2002 had a positive impact on the technological complexity of manufacturing exports. These findings remain robust, supporting the hypothesis that the policy shock associated with administrative approval reform contributed to an increase in the technological complexity of manufacturing exports.

**Table 3. Robustness test: replacement of independent variables and single year test.**

| VARIABLES | (1) | (2) | (3) |
|---|---|---|---|
| | lfirmprody | lfirmprody | lfirmprody |
| alc_department | 0.0119*** | | |
| | (0.0042) | | |
| lalc_item | | 0.0100*** | |
| | | (0.0007) | |
| 2002treat×alc | | | 0.0174* |
| | | | (0.0096) |
| Constant | 5.7162*** | 4.3286*** | 5.6390*** |
| | (0.0636) | (0.0429) | (0.1012) |
| Controls | YES | YES | YES |
| Year FE | YES | YES | YES |
| Firm FE | YES | YES | YES |
| Observations | 542,650 | 542,650 | 207,379 |
| R² | 0.7792 | 0.7702 | 0.7701 |

(3) Parallel trend test

The results of the parallel trend test show that before the implementation of the administrative approval reform, there was no significant difference in the technological complexity of manufacturing exports between cities with and without administrative approval centers (Fig 3). This suggests that the estimation results are robust and that any observed effects can be attributed to the reform rather than pre-existing differences between the cities. Additionally, we observe a lagged effect of administrative approval reform on the technological complexity of manufacturing exports, with a significant positive impact emerging starting from the second year after the establishment of administrative approval centers. The lack of significance in the first year may be due to the time required for enterprises to adjust to the new policy environment. During the early stages of reform, enterprises need time to acclimate to the changes in operational procedures, administrative processes, and regulatory requirements. Although the reform aims to reduce institutional transaction costs, enterprises may experience a temporary decline in export efficiency during this adaptation period, which could negatively impact the complexity of their export technologies. Furthermore, the estimated coefficients begin to decline from the fourth year (Fig 3), indicating a weakening of the reform's positive effects on the technological complexity of manufacturing exports. This pattern may reflect the gradual adaptation of enterprises, where the initial gains from the reform diminish over time as firms fully adjust and begin to capitalize on the policy benefits. Eventually, the marginal benefits of the policy decrease, leading to a plateau in the reform's impact and a reduction in the technological complexity of manufacturing exports during the fourth year.

(4) Placebo test

Although we introduced control variables and fixed effects in the baseline model to mitigate potential issues related to omitted variables, some time- and location-varying factors may still be difficult to control completely. Therefore, following the methodology of Zhou et al. (2018) [32], we conducted a test to assess whether the omission of variables resulted in biased estimation results. Specifically, according to Equation (13), the expression for the estimated coefficient $\hat{\beta}$ is as follows:

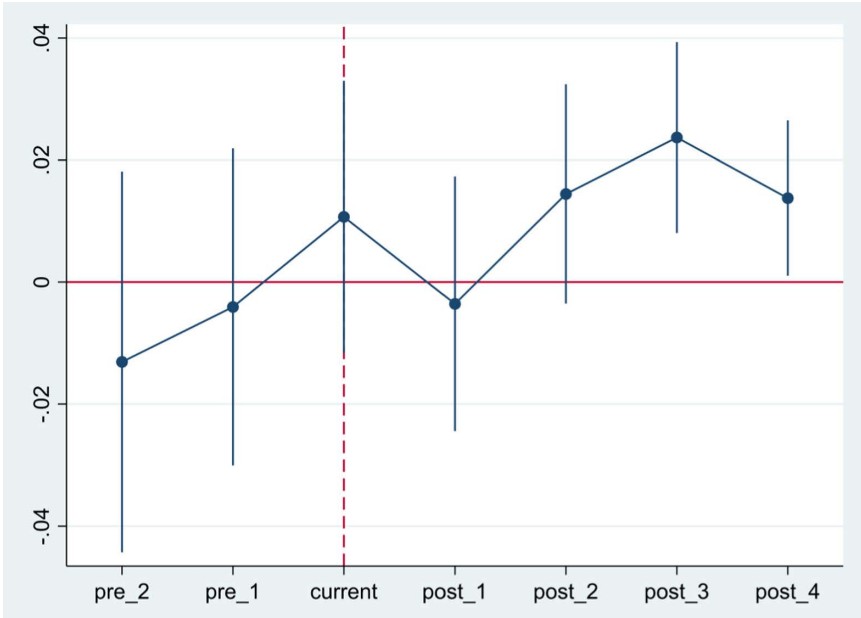

**Fig 3. Parallel trend test.** The parallel trend test is an important prerequisite for validating causal relationships in the study. This figure visually presents whether the treatment group and control group exhibited similar trends before the policy implementation, helping us assess whether the research design meets the underlying assumptions.

$$\hat{\beta} = \beta + \gamma \frac{cov\left(alc\_dum, \varepsilon / z\right)}{var\left(alc\_dum / z\right)} \tag{18}$$

Here, Z represents the control variables. If $\gamma$ =0, unobserved factors do not interfere with the estimation results, and the coefficient estimation *alc_dum* is unbiased. Since a direct test for $\hat{\beta}$ is not feasible, if a variable could replace *alc_dum* and theoretically have no impact on the technological complexity of manufacturing exports, it implies that the true value of $\beta$ is 0. If it is possible to estimate $\hat{\beta}$ as 0 under such conditions, it can be inferred that $\gamma$ =0,. To address this, we introduced randomness to the impact of administrative approval centers on specific regions and repeated this random process 100 times. This ensures that administrative approval centers do not influence the technological complexity of manufacturing exports ($\hat{\beta}$ =0), and it allows the estimation of $\hat{\beta}_{random}$. The findings indicate that $\hat{\beta}_{random}$ is concentrated around 0, allowing us to infer that $\gamma$ =0 (Fig 4). This validates that the estimation results for the impact of administrative approval reform on the technological complexity of manufacturing exports are not affected by unobservable factors, demonstrating the robustness of the empirical results.

(5) Other policy shocks

From 2001 to 2013, China implemented several significant policies, including state-owned enterprise reforms and accession to the World Trade Organization (WTO), in addition to administrative approval reform. The timing of these policies closely coincided with the administrative approval reform, which could potentially lead to estimation biases. To address the issue of omitted variable bias, we separately controlled for the regional proportion of state-owned enterprises and the dependence on foreign trade, which serve as proxies for the

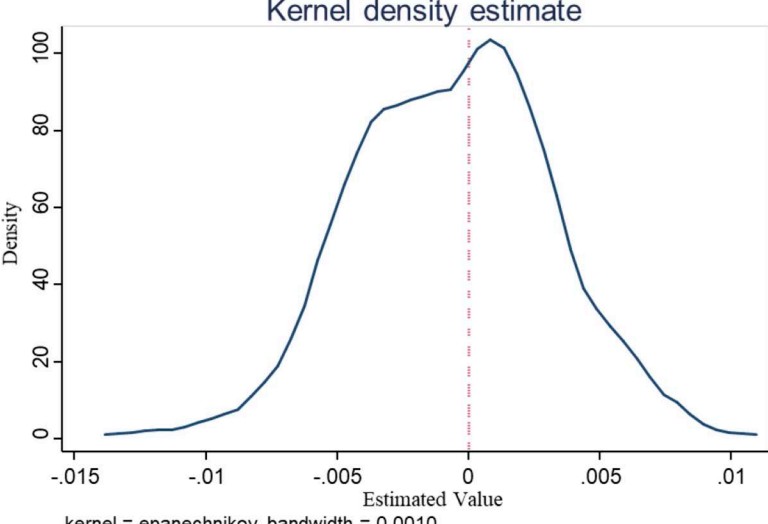

**Fig 4. Placebo test.** The placebo test is used to assess the robustness and reliability of the research findings by simulating a scenario where no true causal relationship exists and observing whether the results remain significant. This figure provides crucial evidence for determining whether the core conclusions of the study are influenced by other confounding factors.

state-owned enterprise reform and the impact of WTO accession, respectively. The results are presented in Table 4, where columns (1) to (3) show the regression results with controls for these other major policies. Regardless of whether we consider the time dimension or the intensity of the administrative approval reform, the findings consistently indicate that administrative approval reform continues to significantly enhance the technological complexity of manufacturing exports. These results remain robust, suggesting that the positive impact of administrative approval reform is not confounded by the other concurrent policies.

(6) Selection Bias

The data for this study are sourced from the Industrial Enterprise Database and the Customs Database, which contain a substantial number of domestic sales enterprises within the sample. Excluding domestic sales enterprises when calculating the technological complexity of manufacturing exports could introduce sample selection bias. To address this issue, we follow the methodology outlined Su et al. (2018) [29] and employ the Heckman two-step procedure for robustness checks.

Firstly, we utilize the Probit model to estimate the probability of enterprises' engagement in export activities. Specifically, we establish a latent variable model, taking into full account a multitude of critical factors influencing enterprises' export decisions. By doing so, we can determine whether an enterprise has an export inclination. Based on this, the Inverse Mills Ratio (IMR) is derived. Furthermore, to enhance the validity of model estimation, as per the recommendation of Su et al. (2018) [29], at least one variable within the set of control variables of the enterprise's export - participation decision equation should be excluded from the set of control variables in the equation determining the quality of the enterprise's export products. Following the approach of existing literature, this paper selects the enterprise's lagged - one - period export dummy variable as the exclusion variable. Subsequently, we incorporate the obtained IMR into the key regression equation and conduct a comprehensive re-examination of the original Equation (13). If the estimated coefficient of the IMR is significantly positive, it indicates the presence of sample selection bias.

**Table 4. Robustness test: Other policy shocks and sample selection bias.**

| VARIABLES | (1) | (2) | (3) | (4) | (5) | (6) |
|---|---|---|---|---|---|---|
| | lfirmprody | lfirmprody | lfirmprody | lfirmprody | lfirmprody | lfirmprody |
| alc_dum | 0.0035*** | | | 0.0035*** | | |
| | (0.0011) | | | (0.0011) | | |
| alc_department | | 0.0125*** | | | 0.0125*** | |
| | | (0.0042) | | | (0.0042) | |
| alc_item | | | 0.0080*** | | | 0.0080*** |
| | | | (0.0008) | | | (0.0008) |
| IMR | | | | -0.0002 | -0.0002 | -0.0020 |
| | | | | (0.0013) | (0.0013) | (0.0013) |
| the reform of state-owned enterprises | -0.0453*** | -0.0451*** | -0.1455*** | -0.0452*** | -0.0450*** | -0.1455*** |
| | (0.0087) | (0.0087) | (0.0078) | (0.0087) | (0.0087) | (0.0078) |
| trade dependence | 0.0773*** | 0.0772*** | 0.2508*** | 0.0774*** | 0.0772*** | 0.2508*** |
| | (0.0104) | (0.0104) | (0.0099) | (0.0104) | (0.0104) | (0.0099) |
| Constant | 5.6881*** | 5.6892*** | 4.5050*** | 5.6881*** | 5.6891*** | 4.5032*** |
| | (0.0636) | (0.0637) | (0.0439) | (0.0636) | (0.0637) | (0.0440) |
| Controls | YES | YES | YES | YES | YES | YES |
| Year FE | YES | YES | YES | YES | YES | YES |
| Firm FE | YES | YES | YES | YES | YES | YES |
| Observations | 542,650 | 542,650 | 542,650 | 542,650 | 542,650 | 542,650 |
| $R^2$ | 0.7793 | 0.7793 | 0.7714 | 0.7793 | 0.7793 | 0.7714 |

We define a firm's participation in exports based on whether its export delivery value exceeds zero in a given year. If the export delivery value is greater than zero, the firm is classified as an exporter; otherwise, it is classified as a non-exporter. The results presented in Table 4, Column (4), show that the estimated coefficient of the inverse Mills ratio (IMR) is not statistically significant, and the null hypothesis of a coefficient equal to zero cannot be rejected. This suggests that sample selection bias does not significantly affect the baseline regression results, and thus, the baseline findings remain valid. Even after incorporating the IMR, the estimated coefficient of alc_dum remains significantly positive. Moreover, we conduct a similar analysis for the robustness indicators of administrative approval centers, including the number of departments and approval items established in these centers. The results presented in Columns (5) and (6) indicate that administrative approval reform continues to significantly enhance the technological complexity of manufacturing exports. These findings further confirm the robustness of the baseline regression results.

(7) Endogenous problem

Although this study employs a difference-in-differences approach and controls for various fixed effects and other covariates in the baseline regression to mitigate potential endogeneity concerns, the possibility of estimation bias remains due to the bidirectional causality between the explanatory and dependent variables. According to the underlying logic of institutional innovation diffusion within the Chinese government, when a higher-level government extends a policy from pilot regions to others, it typically provides policy guidance, protection, and support. This often leads to the short-term realization of institutional dividends. In seeking these dividends, local governments actively pursue preferential access to new institutional arrangements from higher-level authorities. In such cases, the higher-level government takes into account the economic and social conditions of various regions, as well as regional

development policies, when deciding the sequencing of policy implementation. Consequently, the diffusion of institutional innovation is not strictly exogenous; rather, the economic level and degree of openness in different regions are crucial internal factors that local governments prioritize when implementing administrative approval reforms [5]. Therefore, bidirectional causality may exist between the establishment of administrative approval centers and the technological complexity of enterprises' exports.

To address potential endogeneity concerns, we adopt an instrumental variable (IV) approach. Following Zhu and Zhang (2015) [35], we hypothesize that the establishment of administrative approval centers may be influenced by the presence of similar centers in other cities within the same province. The greater the number of administrative approval centers established in neighboring cities, the more likely it is that a local government will establish its own. However, the number of administrative approval centers in other cities is unlikely to directly affect the production decisions of local enterprises. Thus, we use the number of administrative approval centers in other cities within the same province from the previous year as an instrument, which satisfies both the exogeneity and relevance conditions for instrumental variables. As shown in Table 5, column (1), after implementing the IV correction, the estimated coefficient of the core explanatory variable, alc_dum, remains significantly positive at the 15% level. Likewise, in columns (2) and (3), where the number of administrative approval centers in other cities is used as an instrument for the number of departments and approval items in the centers, the estimated coefficients for alc_department and alc_item are also significantly positive. These results further affirm that administrative approval reform contributes to enhancing the technological complexity of manufacturing exports, supporting the robustness of our findings.

## 6. Expanding discussion

### 6.1. Heterogeneity analysis

(1) Ownership Structure of Enterprises.

Given that the impact of administrative approval reform may vary across enterprises with different ownership structures, this chapter conducts a regression analysis by ownership type, with the

**Table 5. Endogeneity testing.**

| VARIABLES | (1) | (2) | (3) |
|---|---|---|---|
|  | lfirmprody | lfirmprody | lfirmprody |
| lalc_department | 0.1086* |  |  |
|  | (0.0633) |  |  |
| alc_dum |  | 0.5286* |  |
|  |  | (0.2852) |  |
| lalc_item |  |  | 0.0606* |
|  |  |  | (0.0352) |
| Controls | YES | YES | YES |
| Year FE | YES | YES | YES |
| Firm FE | YES | YES | YES |
| F-value from the first stage | 132.85 | 93.97 | 222.45 |
| Observations | 542,650 | 542,650 | 542,650 |
| $R^2$ | -0.0193 | -0.0336 | -0.0105 |

The first-stage F-test results indicate the effectiveness of the chosen instrumental variables.

**Table 6. Enterprise ownership.**

| VARIABLES | (1) | (2) |
|---|---|---|
| | SOE | non-SOE |
| alc_dum | -0.0006 | 0.0039*** |
| | (0.0026) | (0.0012) |
| Constant | 6.1844*** | 5.6431*** |
| | (0.1660) | (0.0691) |
| Controls | YES | YES |
| Year FE | YES | YES |
| Firm FE | YES | YES |
| Observations | YES | 462,321 |
| R² | YES | 0.7787 |

results summarized in Table 6. Columns (1) and (2) in Table 6 present the regression results for state-owned enterprises (SOEs) and non-state-owned enterprises (non-SOEs), respectively. The coefficient for "alc_dum" in column (1) is not significantly different from zero, indicating that administrative approval reform does not have a substantial effect on the export technological complexity of SOEs. In contrast, in column (2), the coefficient for non-SOEs is 0.0039, significant at the 1% level. This suggests that administrative approval reform has a significant positive impact on non-SOEs, with each 1% increase in the number of administrative approval departments leading to a 0.39% increase in the technological complexity of their exports. This differential effect can be attributed to the distinct roles of SOEs and non-SOEs in China's foreign trade. Non-SOEs, which account for a larger share of both the number of enterprises and trade volume, often face greater challenges in navigating the approval process. Unlike SOEs, non-SOEs lack inherent governmental connections, making them more susceptible to "dual standards" and unequal treatment, which increases institutional transaction costs. Therefore, administrative approval reform, which emphasizes transparency in fee structures and standardized procedures, has a more pronounced impact on non-SOEs by reducing these transaction costs and enhancing their technological complexity in exports.

(2) Urban geographical characteristics

Considering the differences in external trade conditions and economic development levels across cities, the technological complexity of manufacturing exports may exhibit heterogeneous effects. To account for urban heterogeneity, we perform a classified regression analysis, the results of which are presented in Table 7. Columns (1) and (2) report the results for coastal and inland cities, respectively, while columns (3) and (4) present the regression estimates for port and non-port cities. The results show that the estimated coefficients for the explanatory variables in coastal and port cities are significantly positive. This indicates that administrative approval reform has a more pronounced positive impact on the technological complexity of manufacturing exports in coastal cities and port cities compared to inland and non-port cities. Coastal and port cities in China are key hubs for foreign trade activities, and administrative approval reform, as part of the broader institutional preparation for joining international trade organizations, has played a crucial role in reducing institutional transaction costs.

## 6.2. Mechanism analysis

The United States, the European Union, and the OECD have all experienced reforms in their regulatory systems. In the U.S., reforms were implemented through the establishment of a

robust legal framework, the issuance of executive orders, and the creation of regulatory oversight bodies such as the Office of Management and Budget (OMB) and the Office of Information and Regulatory Affairs (OIRA). These measures effectively streamlined government regulatory processes, reduced institutional transaction costs, and improved market efficiency. The EU, on the other hand, standardized regulatory practices across member states through the Impact Assessment Guidelines, which primarily address market and regulatory failures. This effort enhanced regulatory consistency and transparency, promoting better resource allocation and increasing economic competitiveness. Similarly, the OECD improved regulatory efficiency by simplifying administrative procedures and introducing regulatory impact analysis tools such as cost-benefit analyses and risk assessments, thus enhancing the scientific and effective nature of policy decision-making.

These experiences demonstrate that administrative approval reforms play a crucial role not only in enhancing government efficiency and transparency but also in optimizing resource allocation, reducing unnecessary institutional costs, and fostering innovation—thereby driving economic growth and market vitality. Drawing from these countries' experiences, we gain a broader understanding of the potential impact of administrative approval reforms on the technological complexity of manufacturing exports, particularly in promoting technological innovation, increasing export complexity, and enhancing international competitiveness.

Therefore, this study next investigates the impact of administrative approval reforms on innovation. Given that successful innovation and enhanced operational efficiency rely on sustained investments in R&D and fixed assets, this section explores whether administrative approval reform encourages increased R&D expenditure and fixed asset investment, and further assesses its effect on firms' technological innovation. Fixed asset investment is calculated as the difference between the total fixed assets of the current year and the previous year, adjusted for the current year's depreciation. Technological innovation is measured by the output value of new products. The estimated effects of administrative approval reform on R&D expenditure, fixed asset investment, and technological innovation are presented in Columns (3) to (5) of Table 8. The coefficients for alc_dum are positive and statistically significant at the 1% level across all indicators, indicating that administrative approval reform significantly promotes R&D investment, fixed asset investment, and technological innovation.

**Table 7. Urban geographical characteristics.**

| VARIABLES | (1) | (2) | (3) | (4) |
|---|---|---|---|---|
| | coastal cities | inland cities | port cities | non-port cities |
| alc_dum | 0.0035*** | 0.0040* | 0.0037** | 0.0021 |
| | (0.0013) | (0.0021) | (0.0015) | (0.0018) |
| Constant | 5.7443*** | 5.9387*** | 5.7813*** | 5.8417*** |
| | (0.0796) | (0.1659) | (0.0987) | (0.0986) |
| Controls | YES | YES | YES | YES |
| Year FE | YES | YES | YES | YES |
| Firm FE | YES | YES | YES | YES |
| Observations | 365,166 | 177,323 | 286,370 | 190,687 |
| R² | 0.7699 | 0.7944 | 0.7628 | 0.8058 |

## 7. Further discussion based on the perspective of enterprise entry exit

To analyze the impact of administrative approval reform on the sources of increased technological complexity in manufacturing exports, we adopt the methodology of Melitz and Polanec (2015) to decompose the technological complexity of these exports [33]. By identifying the operational status of enterprises, we categorize the sources of improvements in technological complexity into four main effects: enterprise growth, inter-enterprise reallocation, the entry of new enterprises, and the exit of existing enterprises. The decomposition results are summarized in Formula (18):

$$\Delta prody = \underbrace{\Delta \overline{prody}_S}_{\text{Corporate growth effect}} + \underbrace{\Delta \text{cov}_S}_{\text{reallocation effect between enterprises}}$$
$$+ \underbrace{S_{E2}\left(prody_{E2} - prody_{S2}\right)}_{\text{Entry effect}} - \underbrace{S_{X1}\left(prody_{X1} - prody_{S1}\right)}_{\text{Exit effect}} \quad (18)$$

Equation (18) delineates the sources contributing to the enhancement of technological complexity in manufacturing exports from Period 1 to Period 2. The first through fourth terms denote the effects of enterprise growth, inter-enterprise reallocation, the entry of new enterprises, and the exit of existing enterprises, respectively. Subscripts $E$, $X$, and $S$ represent entering enterprises, exiting enterprises, and surviving enterprises, while $S_{it}$ represents the export share of enterprise i at time t.

The enterprise growth effect represents the average change in the technological complexity of manufacturing exports among surviving firms. It captures the contribution of growth within existing manufacturing enterprises to the overall variation in product technological complexity. Specifically, this effect reflects how the expansion of established firms drives improvements in the technological sophistication of the products they export. The inter-enterprise reallocation effect, on the other hand, measures the covariance between technological complexity in manufacturing exports and market share among surviving enterprises. This effect underscores the role of resource reallocation, which arises from shifts in market shares across firms. A positive inter-enterprise reallocation effect suggests that resources are being reallocated from firms with lower technological complexity to those with higher technological complexity. Such a reallocation enhances the aggregate technological complexity of the sector, as it facilitates the growth of more technologically advanced firms at the expense of less advanced ones.

Table 8. Mechanism verification.

| VARIABLES | (1) | (2) | (3) | (4) | (5) |
|---|---|---|---|---|---|
| | ITC | PITC | R&D | FAI | INV |
| alc_dum | -0.0214** | -0.0194* | 0.0568*** | 0.0256*** | 0.0699*** |
| | (0.0093) | (0.0107) | (0.0054) | (0.0050) | (0.0052) |
| Constant | 6.6270*** | 6.3105*** | -1.2307*** | 6.3728*** | 0.8047*** |
| | (0.2126) | (0.3161) | (0.1639) | (0.3060) | (0.1787) |
| Controls | YES | YES | YES | YES | YES |
| Year FE | YES | YES | YES | YES | YES |
| Firm FE | YES | YES | YES | YES | YES |
| Observations | 477,134 | 477,134 | 104,948 | 166,158 | 267,236 |
| R² | 0.9178 | 0.8925 | 0.6763 | 0.7833 | 0.4789 |

The entry effect refers to the contribution of new enterprises entering the manufacturing export sector to changes in technological complexity. This effect is positive when the technological complexity of entering enterprises exceeds that of surviving enterprises; otherwise, it is negative. The exit effect denotes the contribution of exiting enterprises to changes in technological complexity. This effect is negative when the technological complexity of exiting enterprises exceeds that of surviving enterprises; otherwise, it is positive. Griliches and Regev (1995) and Su et al. (2018) define the second term as the inter-enterprise reallocation effect, the third term as the entry effect [29,34] and the fourth term as the exit effect, collectively referred to as the resource allocation effect. This terminology is adopted in the present study.

Table 9 presents the overall decomposition results for the changes in the technological complexity of manufacturing exports in Column (3). The analysis reveals that, during the sample period, the annual average growth rate of technological complexity in manufacturing exports is 0.1. Notably, the largest contribution to this growth comes from the enterprise growth effect among surviving enterprises, which accounts for 71% of the total change. This is followed by the exit effect, which contributes 61%. In contrast, both the inter-enterprise reallocation effect and the entry effect are negative. This suggests inefficiencies in resource allocation among surviving enterprises, as resources do not effectively flow from firms with lower technological complexity to those with higher complexity. Furthermore, the products of newly entering enterprises exhibit a lower level of technological complexity compared to those of surviving firms, indicating that the entry of new enterprises does not significantly contribute to the overall increase in export technological complexity.

To evaluate the quality of administrative approval reforms across different regions, we classified cities into two categories: those with administrative approval centers and those without. We then decomposed the technological complexity of manufacturing exports for each category. The results are presented in Columns (4) and (5) of Table 9. The findings reveal that, regardless of city type, the technological complexity of manufacturing exports is consistently increasing, with the largest contribution coming from the growth effects of existing enterprises, followed by the exit effects. However, when comparing the two city types, it is evident that cities with more favorable administrative approval reforms show higher values across all components, except for the growth effects of surviving enterprises. Notably, in cities with less effective administrative approval reforms, both the reconfiguration effects and resource allocation effects among existing enterprises are negative. In contrast, in cities with more effective reforms, these effects are positive. This suggests that the positive impact of improved administrative approval reforms on the technological complexity of manufacturing exports is primarily channeled through these two mechanisms: better resource reallocation and enterprise reconfiguration.

**Table 9. Decomposition results of the sources of changes in the complexity of export technology in the manufacturing industry.**

| (1) | (2) | (3) | (4) | (5) |
|---|---|---|---|---|
| variable | Variable Declaration | Total changes | Cities without established administrative approval centers | Administrative Approval Center City |
|  | Total changes | 0.1 | 0.032 | 0.129 |
| diff_within | Corporate growth effect | 0.071 | 0.085 | 0.065 |
| diff_without | reallocation effect between enterprises | -0.001 | -0.01 | 0.004 |
| entan_efft | Entry effect | -0.035 | -0.059 | -0.026 |
| exit_efft | Exit effect | 0.061 | 0.053 | 0.064 |
| allocate | Resource allocation effect | 0.029 | -0.053 | 0.042 |

Based on the preceding analysis, we have preliminarily identified the impact of administrative approval reform on the various sources of technological complexity in manufacturing exports. To further investigate the influence of administrative approval reform on the variation in export technological complexity from different sources across cities, we construct the following model:

$$lnY_{jt} = \beta_0 + \beta_1 ALC_{jt} + \beta Z_{jt} + v_j + \mu_{jt} \tag{19}$$

In this equation, $Y_{jt}$ represents the various components of changes in technological complexity in manufacturing exports obtained through decomposition in the city $j$ in the year $t$, with the remaining variables consistent with Equation (13).

Table 10 presents the regression results examining the impact of administrative approval reform on changes in the overall technological complexity of urban manufacturing exports. Columns (1) to (5) report the regression estimates of administrative approval reform on the enterprise self-growth effect, inter-enterprise resource allocation effect, entry effect, exit effect, and redistribution effect, respectively. According to the results in column (5), the estimated coefficient of the explanatory variable "alc_dum" is significant at the 10% level, indicating that the resource allocation effect is the primary channel through which administrative approval reform enhances the technological complexity of manufacturing exports. Specifically, the reform facilitates the redistribution of market share from enterprises with lower export technological complexity to those with higher technological complexity, thus contributing to the overall increase in export technological complexity across cities. Furthermore, the positive and significant result in column (3) suggests that the entry effect also plays an important role in improving the technological complexity of urban manufacturing exports.

To ensure the robustness of the conclusions, this chapter further examines the effect of administrative approval reform from two dimensions: the number of administrative approval departments and the number of approval items. Columns (6) to (10) and columns (11) to (15) of Table 10 report the impacts of the number of administrative approval departments and the number of approval items on the sources of changes in the overall technological complexity of cities, respectively. The regression result in column (15) indicates that administrative approval reform facilitates internal resource redistribution within cities, which is consistent with the finding in column (5).

## 8. Conclusion and policy recommendations

Administrative approval reform can reduce institutional transaction costs and unlock institutional dividends through systematic mechanisms, thereby enhancing the technological complexity of manufacturing exports. However, existing literature provides limited insights into the specific relationship between administrative approval reform and the technological complexity of manufacturing exports. This paper treats administrative approval system reform as a policy shock and examines its impact and underlying mechanisms on the technological complexity of manufacturing exports from both theoretical and empirical perspectives. Theoretical model analysis suggests that by reducing both marginal and fixed costs for manufacturing firms in adjusting product complexity, administrative approval reform effectively promotes the enhancement of technological complexity in manufacturing exports. Empirical analysis confirms that administrative approval reform significantly increases the technological complexity of manufacturing exports, and this conclusion remains robust after a series of robustness checks. Micro-level mechanism analysis reveals that reducing institutional transaction costs is a key channel through which administrative approval reform exerts its effects. Additionally, this study demonstrates that increasing R&D investment, fixed asset investment,

**Table 10. The impact of administrative approval reform on the sources of changes in manufacturing export technology complexity.**

| | (1) | (2) | (3) | (4) | (5) |
|---|---|---|---|---|---|
| | diff_within | diff_without | entan_efft | exit_efft | allocate |
| alc_dum | -0.0125 | 0.0142 | 0.1381* | 0.0399 | 0.0589* |
| | (0.0747) | (0.0505) | (0.0729) | (0.0751) | (0.0357) |
| Constant | 0.2810 | 0.2321 | -0.6171 | -0.4478 | 0.0505 |
| | (1.0959) | (0.7433) | (1.1453) | (1.0969) | (1.9789) |
| Controls | YES | YES | YES | YES | YES |
| City FE | YES | YES | YES | YES | YES |
| Year FE | YES | YES | YES | YES | YES |
| Observations | 986 | 1,088 | 1,048 | 867 | 814 |
| $R^2$ | 0.1672 | 0.1187 | 0.3348 | 0.5243 | 0.2191 |
| | (6) | (7) | (8) | (9) | (10) |
| | diff_within | diff_without | entan_efft | exit_efft | allocate |
| lalc_department | -0.0012 | 0.0038 | 0.0383* | 0.0154 | 0.2000 |
| | (0.0201) | (0.0136) | (0.0196) | (0.0200) | (0.1335) |
| Constant | 0.2773 | 0.2341 | -0.6048 | -0.4391 | 0.0797 |
| | (1.0957) | (0.7432) | (1.1450) | (1.0967) | (1.9785) |
| Controls | YES | YES | YES | YES | YES |
| City FE | YES | YES | YES | YES | YES |
| Year FE | YES | YES | YES | YES | YES |
| Observations | 986 | 1,088 | 1,048 | 867 | 814 |
| $R^2$ | 0.1672 | 0.1187 | 0.3350 | 0.5246 | 0.2198 |
| | (11) | (12) | (13) | (14) | (15) |
| | diff_within | diff_without | entan_efft | exit_efft | allocate |
| lalc_item | 0.0020 | 0.0014 | 0.0278** | 0.0114 | 0.0401* |
| | (0.0132) | (0.0090) | (0.0132) | (0.0132) | (0.0235) |
| Constant | 0.2721 | 0.2351 | -0.5769 | -0.4419 | 0.0645 |
| | (1.0957) | (0.7433) | (1.1442) | (1.0965) | (1.9778) |
| Controls | YES | YES | YES | YES | YES |
| City FE | YES | YES | YES | YES | YES |
| Year FE | YES | YES | YES | YES | YES |
| Observations | 986 | 1,088 | 1,048 | 867 | 814 |
| $R^2$ | 0.1672 | 0.1186 | 0.3355 | 0.5247 | 0.2200 |

and fostering technological innovation in export-oriented enterprises are critical pathways through which administrative approval reform influences the technological complexity of manufacturing exports.

Heterogeneity analysis reveals that the effects of administrative approval reform on enhancing the technological complexity of manufacturing exports are particularly pronounced for non-state-owned enterprises, as well as those located in coastal and port cities. Additionally, through the decomposition of overall city-level export technological complexity (DOP decomposition), this study identifies inter-firm resource allocation as the primary mechanism through which administrative approval reform drives improvements in technological complexity at the city level. Specifically, administrative approval reform facilitates the reallocation of market share from firms with lower technological complexity to those with higher technological complexity, thereby enhancing the overall export technological complexity of cities.

Based on the research conclusions of this article, the following three policy implications can be derived:

First, further reduce institutional transaction costs: Administrative approval reform has significantly reduced institutional transaction costs by shortening approval times and lowering expenses for enterprises. It is recommended that the government continue to streamline approval processes and increase transparency, particularly for non-state-owned enterprises. Simplifying approval procedures and enhancing operational efficiency will further improve the technological complexity of exports.

Second, optimize resource allocation to support the development of high-tech complexity enterprises: Policies should prioritize optimizing resource allocation among enterprises through market-based mechanisms, fostering the transfer of market share to firms with strong innovation capabilities and higher export technological complexity. By enhancing market competition mechanisms, high-tech enterprises will be better positioned to access market opportunities, thereby contributing to the overall increase in export technological complexity at the city level.

Third, strengthen R&D support policies to drive technological innovation: Given that R&D investment and technological innovation are key drivers of increased export technological complexity, the government should continue to enhance support for enterprise R&D activities, such as through tax incentives and R&D subsidies. This approach will encourage firms to reinvest the savings into innovation, driving technological upgrades in export products and improving their competitiveness in the international market.

This paper does have certain limitations. Firstly, due to data constraints, it relies on data from 2001 to 2013, limiting its ability to conduct a long-term dynamic analysis of the impact of administrative approval reform on the technological complexity of manufacturing exports. Secondly, this study focuses primarily on the effect of the administrative approval system on export technological complexity. In the context of economic globalization, exploring the role of institutional optimization in enhancing positions within global value chains presents a valuable direction for future research. Future research could focus on two key areas: First, leveraging long-term data through methods such as regression discontinuity design (RDD) or dynamic panel data models to analyze the long-term dynamic effects of administrative approval reform on the evolution of technological complexity in manufacturing exports. Second, employing difference-in-differences (DID) methods to explore the interaction between administrative approval reform and global value chain restructuring. This would allow for a deeper understanding of how administrative approval reform impacts export technological complexity across various segments, while clarifying the role of institutional optimization in advancing global value chain positions.

## Author contributions

**Data curation:** Tingting Jiang.

**Formal analysis:** Tingting Jiang.

**Funding acquisition:** Tingting Jiang.

**Investigation:** Tingting Jiang.

**Methodology:** Tingting Jiang.

**Project administration:** Tingting Jiang.

**Resources:** Tingting Jiang.

**Software:** Tingting Jiang.

**Supervision:** Zhibin Zhang.

**Validation:** Zhibin Zhang.

**Visualization:** Tingting Jiang.

**Writing – original draft:** Tingting Jiang.

**Writing – review & editing:** Zhibin Zhang.

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
