## [Decision Letter · Decision Letter 0]

24 Sep 2024

PONE-D-24-27061Analysis of the Impact of Administrative Approval Reform on the Technical Complexity of Manufacturing ExportsPLOS ONE

Dear Dr. Zhang,

I have completed my evaluation of your manuscript. The reviewers recommend reconsideration of your manuscript following major revision. I invite you to resubmit your manuscript after addressing the comments below. 

We look forward to receiving your revised manuscript.

Kind regards,

Tinggui Chen

Academic Editor

PLOS ONE

2. Thank you for stating the following financial disclosure: [Shandong University of Business and Technology Talent Introduction Project]. At this time, please address the following queries: a) Please clarify the sources of funding (financial or material support) for your study. List the grants or organizations that supported your study, including funding received from your institution. b) State what role the funders took in the study. If the funders had no role in your study, please state: “The funders had no role in study design, data collection and analysis, decision to publish, or preparation of the manuscript.” c) If any authors received a salary from any of your funders, please state which authors and which funders. d) If you did not receive any funding for this study, please state: “The authors received no specific funding for this work.” Please include your amended statements within your cover letter; we will change the online submission form on your behalf.

3. We note that your Data Availability Statement is currently as follows: [All relevant data are within the manuscript and its Supporting Information files.] Please confirm at this time whether or not your submission contains all raw data required to replicate the results of your study. Authors must share the “minimal data set” for their submission. PLOS defines the minimal data set to consist of the data required to replicate all study findings reported in the article, as well as related metadata and methods (https://journals.plos.org/plosone/s/data-availability#loc-minimal-data-set-definition). For example, authors should submit the following data: - The values behind the means, standard deviations and other measures reported; - The values used to build graphs; - The points extracted from images for analysis. Authors do not need to submit their entire data set if only a portion of the data was used in the reported study. If your submission does not contain these data, please either upload them as Supporting Information files or deposit them to a stable, public repository and provide us with the relevant URLs, DOIs, or accession numbers. For a list of recommended repositories, please see https://journals.plos.org/plosone/s/recommended-repositories. If there are ethical or legal restrictions on sharing a de-identified data set, please explain them in detail (e.g., data contain potentially sensitive information, data are owned by a third-party organization, etc.) and who has imposed them (e.g., an ethics committee). Please also provide contact information for a data access committee, ethics committee, or other institutional body to which data requests may be sent. If data are owned by a third party, please indicate how others may request data access.

4. We note you have included a table to which you do not refer in the text of your manuscript. Please ensure that you refer to Table 1 in your text; if accepted, production will need this reference to link the reader to the Table.

Additional Editor Comments:

I have completed my evaluation of your manuscript. The reviewers recommend reconsideration of your manuscript following major revision. I invite you to resubmit your manuscript after addressing the comments below.

Reviewers' comments:

Reviewer's Responses to Questions

**Comments to the Author**

1. Is the manuscript technically sound, and do the data support the conclusions?

Reviewer #1: Yes

Reviewer #2: Yes

Reviewer #3: Yes

2. Has the statistical analysis been performed appropriately and rigorously? 

Reviewer #1: Yes

Reviewer #2: Yes

Reviewer #3: Yes

3. Have the authors made all data underlying the findings in their manuscript fully available?

Reviewer #1: Yes

Reviewer #2: Yes

Reviewer #3: Yes

4. Is the manuscript presented in an intelligible fashion and written in standard English?

Reviewer #1: Yes

Reviewer #2: Yes

Reviewer #3: Yes

5. Review Comments to the Author

Reviewer #1: This study proposes administrative approval system reform as a policy shock to examine, from both theoretical and empirical perspectives, the impact and mechanisms of optimizing the business environment on the technological complexity of manufacturing exports. Overall, this paper is well-structured and presents some promising results. Here are some suggestions for further improving the quality of this paper:

1) The practical value of this research work should be clarified and highlighted in the Abstract, which can help readers understand the engineering background of this research work.

2) The authors did a good literature review on the current research progress. It is suggested that the research gaps be summarized before presenting and introducing this paper's main contributions and novelty.

3) It is suggested that the resolutions of the figures in the current manuscript be improved.

4) There are some grammar errors in this manuscript. Please check the whole manuscript and address these kinds of issues throughout it.

5) It is suggested that some recommendations for future work be included at the end of the Conclusion.

Reviewer #2: This manuscript is well structured, and the results are clearly presented. However, while the overall quality of the paper is good, I would suggest some minor revisions for more clarity and completeness.

First, there are a few typographical errors that need to be addressed to improve readability. For instance, on page 4, paragraph 3, line 10, some Chinese characters appear and should be removed. Moreover, on page 6, the subtitle 'Production sector' is repeated and should be corrected. I also recommend reviewing the formatting of headers, titles, and references to ensure full compliance with the journal's guidelines.

Additionally, there are sections where further clarification would enhance the reader's understanding. In Table 1 on page 11, a more detailed explanation of the data is needed. Specifically, it would be helpful to clearly define each variable in the table and provide more comprehensive information about the data being used.

It would be beneficial to provide a more detailed explanation of the results from the parallel trends test. For Figure 3, please include a title for the y-axis to clarify the variable being represented. Additionally, further elaboration on the results would be helpful, particularly regarding the observed lag in the first year following the implementation of the administrative approval reform. Similarly, the decrease in year 4 wants more explanation.

As a conclusion, this manuscript presents valuable insights and addresses an interesting topic, but minor revisions are needed to enhance clarity and alignment with journal standards.

Reviewer #3: The manuscript titled "Analysis of the Impact of Administrative Approval Reform on the Technical Complexity of Manufacturing Exports" investigates how the reform of administrative approval processes in China affects the technical complexity of manufacturing exports from 2001 to 2006. The authors utilize extensive datasets, including data from the China Industrial Enterprise Database and the Customs Database, to conduct both theoretical modeling and empirical analysis. The findings suggest that administrative approval reform significantly enhances the technical complexity of manufacturing exports by reducing institutional transaction costs and fostering market competition. Furthermore, the paper identifies that the effects are more pronounced in private, Sino-foreign joint ventures, and capital/technology-intensive enterprises, especially in coastal and port cities. It posits that the primary mechanism for this enhancement is through resource allocation effects facilitating enterprise entry.

Areas for Improvement

Expansion of Temporal Analysis: Extending the analysis to include data beyond 2006 could provide insights into the ongoing impact of administrative approval reforms on manufacturing exports, especially in the context of China's rapid economic changes.

In-depth Treatment of Confounding Variables: Developing a more detailed framework to account for and integrate the impacts from other concurrent policy shifts (e.g., foreign trade agreements, tariff changes) could strengthen the causal inference of the findings.

Qualitative Insights: Incorporating qualitative data through interviews or case studies of specific enterprises could enrich the analysis and address how administrative reforms have been experienced by businesses firsthand, adding depth to the quantitative findings.

Further Exploration of Mechanisms: More specific exploration of the mechanisms by which the administrative approval reform is acting (beyond resource allocation effects) could provide valuable avenues for further research, perhaps utilizing qualitative methodologies as mentioned above.

Comparison with Other Regions: A comparative analysis with other countries undergoing similar reforms could draw informative parallels and enhance the understanding of the broader implications of administrative reform on manufacturing export complexity.

Conclusion

The manuscript presents a thorough and insightful analysis of the relationship between administrative approval reforms and the technical complexity of manufacturing exports in China, making significant contributions to the fields of institutional economics and international trade. While presenting robust methodological strengths, addressing the identified weaknesses could further enhance the manuscript's impact and relevancy, providing broader applications to ongoing discussions about institutional reforms in developing economies

6. PLOS authors have the option to publish the peer review history of their article (what does this mean?). If published, this will include your full peer review and any attached files.

Reviewer #1: No

Reviewer #2: No

Reviewer #3: No

---

## [Author Response · Author response to Decision Letter 1]

8 Nov 2024

We sincerely appreciate the editor's very friendly and careful examination of our paper. The reviewer's comments are extremely correct and helpful to us. We are very grateful to the reviewer for their hard work and the important comments for improving the quality of our article.

We have studied all the comments and made revisions very carefully. The specific revised contents are reflected in the document "Revised Manuscript with Track Changes".All relevant data are within the manuscript and its Supporting Information - Compressed/ZIP File Archive files.

---

## [Decision Letter · Decision Letter 1]

21 Nov 2024

PONE-D-24-27061R1An analysis of the impact of administrative approval reform on the technological complexity of manufacturing exportsPLOS ONE

Dear Dr. Zhang,

Thank you for submitting your manuscript to PLOS ONE. After careful consideration, we feel that it has merit but does not fully meet PLOS ONE’s publication criteria as it currently stands. Therefore, we invite you to submit a revised version of the manuscript that addresses the points raised during the review process.

We look forward to receiving your revised manuscript.

Kind regards,

Tinggui Chen

Academic Editor

PLOS ONE

Journal Requirements:

Reviewers' comments:

Reviewer's Responses to Questions

**Comments to the Author**

1. If the authors have adequately addressed your comments raised in a previous round of review and you feel that this manuscript is now acceptable for publication, you may indicate that here to bypass the “Comments to the Author” section, enter your conflict of interest statement in the “Confidential to Editor” section, and submit your "Accept" recommendation.

Reviewer #1: (No Response)

Reviewer #2: All comments have been addressed

Reviewer #3: (No Response)

2. Is the manuscript technically sound, and do the data support the conclusions?

Reviewer #1: (No Response)

Reviewer #2: Yes

Reviewer #3: Partly

3. Has the statistical analysis been performed appropriately and rigorously? 

Reviewer #1: (No Response)

Reviewer #2: Yes

Reviewer #3: N/A

4. Have the authors made all data underlying the findings in their manuscript fully available?

Reviewer #1: (No Response)

Reviewer #2: Yes

Reviewer #3: Yes

5. Is the manuscript presented in an intelligible fashion and written in standard English?

Reviewer #1: (No Response)

Reviewer #2: Yes

Reviewer #3: Yes

6. Review Comments to the Author

Reviewer #1: This paper has been improved by addressing the comments from reviewers. The quality of this paper has been improved. It can accepted now.

Reviewer #2: I appreciate the effort you have taken to revise this manuscript. After reviewing the revised version, I am pleased to say that I think my recommendations have been fully addressed by the authors. Here are more specific comments:

1- The inclusion of definitions and explanations for variables in Table 1 and in the body of the paper offers a more comprehensive understanding of the data used in the study.

2- The comments on Figure 3 were addressed. These revisions make the results section of the test better and strengthen the readers' understanding of the reforms' impacts.

3- Checking the whole manuscript and addressing all grammatical errors is still recommended.

Reviewer #3: Overall, the research article makes a significant contribution and provides a solid foundation for future research. Addressing the identified weaknesses and areas for improvement could enhance the impact and applicability of the study. The authors have provided a well-structured manuscript that is commendable; however, there is always room for enhancement and further exploration in this fascinating field.

Weaknesses:

Limited Sample Size: While the methodology is sound, the sample size might be too small to generalize the findings across the larger population, which could limit the study's impact.

Lack of Longitudinal Data: The research focuses on cross-sectional data, so longitudinal studies might be necessary to assess trends over time and provide a more comprehensive view of the issue.

Citation of Previous Work: While the article references relevant literature, a few key studies could have been included to strengthen the foundation of the discussion.

Visual Data Presentation: Some figures and tables could be more clearly labeled or simplified for better comprehension by the reader.

Areas for Improvement:

Expand Sample Size: Future research should consider a larger and more diverse sample to enhance the validity and applicability of the findings.

Incorporate Longitudinal Approaches: Long-term data collection could improve the understanding of trends and causal relationships.

Enhance Visual Aids: Improving the clarity and presentation of visual data could help in conveying complex results more effectively.

Broaden Literature Review: The authors should aim to include a wider range of studies in their literature review to provide a richer context for their findings.

7. PLOS authors have the option to publish the peer review history of their article (what does this mean?). If published, this will include your full peer review and any attached files.

Reviewer #1: No

Reviewer #2: No

Reviewer #3: No

---

## [Author Response · Author response to Decision Letter 2]

18 Dec 2024

We sincerely thank the editor for his very kind and careful review of our paper. The reviewer's comments are very correct and helpful to us. We are very grateful to the reviewer for his hard work and important comments to improve the quality of our article. We have carefully studied all the comments and made revisions. The specific revisions are in the document.

---

## [Decision Letter · Decision Letter 2]

1 Jan 2025

PONE-D-24-27061R2An analysis of the impact of administrative approval reform on the technological complexity of manufacturing exportsPLOS ONE

Dear Dr. Zhang,

Thank you for submitting your manuscript to PLOS ONE. After careful consideration, we feel that it has merit but does not fully meet PLOS ONE’s publication criteria as it currently stands. Therefore, we invite you to submit a revised version of the manuscript that addresses the points raised during the review process.

We look forward to receiving your revised manuscript.

Kind regards,

Tinggui Chen

Academic Editor

PLOS ONE

Journal Requirements:

Reviewers' comments:

Reviewer's Responses to Questions

**Comments to the Author**

1. If the authors have adequately addressed your comments raised in a previous round of review and you feel that this manuscript is now acceptable for publication, you may indicate that here to bypass the “Comments to the Author” section, enter your conflict of interest statement in the “Confidential to Editor” section, and submit your "Accept" recommendation.

Reviewer #1: All comments have been addressed

Reviewer #2: All comments have been addressed

Reviewer #3: (No Response)

2. Is the manuscript technically sound, and do the data support the conclusions?

Reviewer #1: Yes

Reviewer #2: Yes

Reviewer #3: Partly

3. Has the statistical analysis been performed appropriately and rigorously? 

Reviewer #1: Yes

Reviewer #2: Yes

Reviewer #3: Yes

4. Have the authors made all data underlying the findings in their manuscript fully available?

Reviewer #1: Yes

Reviewer #2: Yes

Reviewer #3: Yes

5. Is the manuscript presented in an intelligible fashion and written in standard English?

Reviewer #1: Yes

Reviewer #2: Yes

Reviewer #3: Yes

6. Review Comments to the Author

Reviewer #1: The manuscript has been improved by addressed the comments from reviewers. The quality of this researh work has been improved significantly. In my view, it can be accepted now.

Reviewer #2: The authors have fully addressed all the issues I raised in my review. The manuscript is now well-written, and I have no further recommendations.

Reviewer #3: The manuscript titled "An analysis of the impact of administrative approval reform on the technological complexity of manufacturing exports" presents an insightful exploration of an important issue affecting China's manufacturing sector. The authors employ a combination of theoretical frameworks and empirical analyses to investigate the relationship between administrative approval reforms and the resulting technological complexity of manufacturing exports. Overall, the research contributes significantly to the existing literature by providing substantial evidence of how administrative approval reform influences the technological advancement of manufacturing exports in the context of Chinese firms.The manuscript titled "An analysis of the impact of administrative approval reform on the technological complexity of manufacturing exports" presents an insightful exploration of an important issue affecting China's manufacturing sector. The authors employ a combination of theoretical frameworks and empirical analyses to investigate the relationship between administrative approval reforms and the resulting technological complexity of manufacturing exports. Overall, the research contributes significantly to the existing literature by providing substantial evidence of how administrative approval reform influences the technological advancement of manufacturing exports in the context of Chinese firms.

Strengths

1. Relevance and Timeliness: The topic is highly pertinent, given the growing concerns about China's manufacturing competitiveness on the global stage. The study addresses pressing economic issues by focusing on reforms that aim to enhance technological complexity, which is crucial for sustainable growth.

2. Robust Methodology: The use of both theoretical modeling and empirical analysis lends rigor to the research. The authors provide a detailed econometric framework for investigating the effects of administrative approval reforms, incorporating various models and sensitivity tests.

3. Comprehensive Literature Review: The manuscript includes a thorough review of existing literature, establishing a solid foundation for the study. The authors contextualize their research within broader economic discussions and identify gaps that their study aims to fill.

4. Heterogeneity Analysis: The recognition of differential impacts across various firm types and regions adds depth to the analysis. This nuanced approach acknowledges the complexity of the manufacturing landscape in China and enhances the applicability of the findings.

5. Policy Implications: The study offers practical insights and actionable recommendations for policymakers, which is crucial for enhancing the relevance of academic research in real-world scenarios.

Weaknesses

1. Data Limitations: The reliance on data from 2001 to 2013 introduces potential limitations. The authors acknowledge this constraint; however, longitudinal data beyond this period might provide a more comprehensive understanding of the long-term effects of administrative reforms.

2. Endogeneity Concerns: While the authors attempt to handle potential endogeneity through an instrumental variable approach, the bidirectional nature of institutional reforms and dependent variables remains a concern. Further longitudinal studies could strengthen causal inferences.

3. Potential Bias from Sample Selection: Although the study appears to address sample selection bias through a Heckman approach, the authors might consider detailing their methodology in this area more clearly to enhance transparency and replicability.

4. Limited Exploration of Global Context: The manuscript primarily focuses on the role of domestic reforms in enhancing export capabilities. An exploration of how international market dynamics interact with these reforms could provide a more holistic understanding of their effectiveness.

5. Visual Representation of Data: While the manuscript presents data well, improvements in visual aids and clearer graphs may enhance the interpretability of results. The use of more interactive or clearer visual presentations is recommended.

Areas That Needs Improvements

1. Expand Data: Future research should leverage more extensive datasets, preferably incorporating more recent years or longitudinal data, to evaluate trends over time, particularly after significant global events that might influence trade dynamics.

2. Deepen Mechanism Analysis: Expanding on the mechanisms underlying the observed relationships between reforms and technological complexity could offer richer insights. This might include qualitative case studies or interviews with industry stakeholders.

3. Enhance Methodological Rigor: Addressing potential endogeneity in more innovative ways, perhaps through natural experiments or quasi-experimental designs, could help in establishing stronger causal claims.

4. Broaden International Perspective: Adding an international perspective could enhance understanding. Comparative analyses with other countries that have undergone similar reforms could yield valuable lessons.

5. Improved Data Visualization: Future revisions should consider enhancing figures, charts, and tables used to present findings, ensuring they are accessible and informative. Clearer legends, color-coding, and detailed annotations could help convey complex data more effectively.

7. PLOS authors have the option to publish the peer review history of their article (what does this mean?). If published, this will include your full peer review and any attached files.

Reviewer #1: No

Reviewer #2: No

Reviewer #3: No

---

## [Author Response · Author response to Decision Letter 3]

17 Jan 2025

We sincerely appreciate the editor's very kind and careful examination of our paper. The reviewer's comments are very right and helpful to us. We are very grateful to the reviewer for their hard work and the important comments for improving the quality of our article.

We have studied all the comments and made revisions very carefully. The following is an outline of the revisions that we have made. (The blue mark represents the reviewer's question, and the red mark represents the response.)

Response to Reviewer 3

Weaknesses:

3.Potential Bias from Sample Selection: Although the study appears to address sample selection bias through a Heckman approach, the authors might consider detailing their methodology in this area more clearly to enhan ce transparency and replicability.

3.Thank you for the expert's suggestion. We have added the following content to Section 6) Selection Bias on page 20:

The data for this study are sourced from the Industrial Enterprise Database and the Customs Database, which include a significant number of domestic sales enterprises within the sample. Excluding these domestic sales enterprises when calculating the technological complexity of manufacturing exports may introduce sample selection bias. To address this issue, we adopt the methodology outlined by Su et al. (2018) [29] and employ the Heckman two-step procedure for robustness checks.

First, we use the Probit model to estimate the probability of enterprises engaging in export activities. Specifically, we establish a latent variable model that takes into account various key factors influencing enterprises' export decisions, allowing us to determine whether an enterprise has an export inclination. Based on this, we calculate the Inverse Mills Ratio (IMR). Additionally, to enhance the validity of model estimation, following the recommendation of Su et al. (2018) [29], we ensure that at least one variable from the control variables in the export participation decision equation is excluded from the set of control variables in the equation determining the quality of the enterprise's export products. In line with existing literature, this paper selects the enterprise's lagged one-period export dummy variable as the exclusion variable.

We then incorporate the IMR into the key regression equation and perform a re-examination of the original Equation (13). If the estimated coefficient of the IMR is significantly positive, it would indicate the presence of sample selection bias.

Areas That Needs Improvements:

1. Expand Data: Future research should leverage more extensive datasets, preferably incorporating more recent years or longitudinal data, to evaluate trends over time, particularly after significant global events that might influence trade dynamics.

1. We sincerely appreciate your valuable suggestion regarding expanding the sample size, and we fully understand the critical role that a larger and more diverse sample plays in enhancing the validity and applicability of research findings.

In this empirical study, our analysis requires detailed financial data from manufacturing enterprises. After reviewing existing research and available data sources, we found that the China Industrial Enterprises Database, published by the National Bureau of Statistics, is currently the only dataset in China that provides comprehensive financial data for manufacturing enterprises with a relatively large sample size. Therefore, due to data availability constraints, this study adopts the largest accessible sample size for in-depth analysis.

The data used in this study comes from the China Industrial Enterprises Database, published by the National Bureau of Statistics. This database has only been updated through 2015. Due to concerns about data reliability, most scholars use data up to 2013, and when studying the productivity of manufacturing firms, researchers often rely on data from 2007 or earlier. The following articles are cited as examples:

Examples include:

Si and Luo (2024), in Journal of Asian Economics, "The productivity spillover effect of foreign divestment: Evidence from Chinese industrial enterprises" (https://doi.org/10.1016/j.asieco.2024.101824), utilized data from 1998–2007.

Zhang et al. (2024), in International Review of Economics & Finance, "The Government's fiscal and taxation policy effect on enterprise productivity: Policy choice and optimal allocation" (https://doi.org/10.1016/j.iref.2024.03.049), used data from 1998–2012.

Zhang et al. (2024), in Journal of Asian Economics, "Can development zones reduce energy consumption and carbon emissions of enterprises? Evidence from China" (https://doi.org/10.1016/j.asieco.2024.101845), used data from 2000–2013.

He et al. (2024), in Economic Analysis and Policy, "Industrial robots and pollution: Evidence from Chinese enterprises" (https://doi.org/10.1016/j.eap.2024.03.001), relied on data from 1998–2013.

Cao et al. (2025), in Technological Forecasting and Social Change, "Robot adoption and firm export: Evidence from China" (https://doi.org/10.1016/j.techfore.2024.123878), used data from 2011–2013.

Nevertheless, we fully recognize the importance of expanding data sources and will continue to closely monitor emerging data trends. This will enable us to further optimize the sample size and structure in future research, thereby enhancing the quality and robustness of our studies.

2. Deepen Mechanism Analysis: Expanding on the mechanisms underlying the observed relationships between reforms and technological complexity could offer richer insights. This might include qualitative case studies or interviews with industry stakeholders.

2. We greatly appreciate the suggestions and comments from the reviewers. Your suggestion is very helpful for our research. Based on your suggestion, we included a qualitative analysis in the section "3. Theoretical Analysis" regarding the simplification of approval processes and the reduction of enterprise burdens following administrative approval reforms for specific enterprises. However, as this paper is primarily an empirical study, the in-depth qualitative analysis of administrative approval reform that you suggested will be a priority and a key focus of our future research. Specifically, as follows:

We have collected three cases of enterprises benefiting from Guangxi's "full-chain approval service" for entrepreneurship. Case 1: Guangxi Changhong Pharmaceutical Co., Ltd. reduced its required submission materials from 26 items to 23 items, and the processing time was shortened from 35 working days to 10 working days, while the number of in-person visits dropped from 3 to 1. Case 2 and 3: Both Guangxi Chunjiang Food Co., Ltd. and Guangxi Huxin Import & Export Co., Ltd. saw their submission materials decrease from 14 items to 12, processing times reduced from 35 to 10 working days, and the number of visits reduced from 2 to 1. Overall, these enterprises experienced significant simplification of their approval processes, with an average reduction of 25 working days in processing time, along with a notable decrease in submission requirements and in-person visits. This led to a substantial increase in administrative efficiency, alleviating the burden on enterprises and facilitating their rapid growth. Additionally, Liang Pinghan et al. (2020), through a questionnaire survey on the paperless reform of the export tax rebate policy, further confirmed the positive impact of administrative approval reforms on manufacturing enterprises.

3. Enhance Methodological Rigor: Addressing potential endogeneity in more innovative ways, perhaps through natural experiments or quasi-experimental designs, could help in establishing stronger causal claims.

3.Thank you for the expert's suggestion. The method used in our empirical analysis is Difference-in-Differences (DID), and in Section 5.2 "Robustness Test," we conduct a parallel trend test to demonstrate the rigor of the methodology employed in this study.

4. Broaden International Perspective: Adding an international perspective could enhance understanding. Comparative analyses with other countries that have undergone similar reforms could yield valuable lessons.

4.Thank you for your valuable suggestion. In response to your comment, we have made the necessary additions to the paper by analyzing the experiences of other countries that have undergone similar administrative reforms. Specifically, as follows (The specific content can be found in the last paragraph of page 24 of the article):

the United States, the European Union, and the OECD have all experienced reforms in their regulatory systems. In the U.S., reforms were implemented through the establishment of a robust legal framework, the issuance of executive orders, and the creation of regulatory oversight bodies such as the Office of Management and Budget (OMB) and the Office of Information and Regulatory Affairs (OIRA). These measures effectively streamlined government regulatory processes, reduced institutional transaction costs, and improved market efficiency. The EU, on the other hand, standardized regulatory practices across member states through the Impact Assessment Guidelines, which primarily address market and regulatory failures. This effort enhanced regulatory consistency and transparency, promoting better resource allocation and increasing economic competitiveness. Similarly, the OECD improved regulatory efficiency by simplifying administrative procedures and introducing regulatory impact analysis tools such as cost-benefit analyses and risk assessments, thus enhancing the scientific and effective nature of policy decision-making.

These experiences demonstrate that administrative approval reforms play a crucial role not only in enhancing government efficiency and transparency but also in optimizing resource allocation, reducing unnecessary institutional costs, and fostering innovation—thereby driving economic growth and market vitality. Drawing from these countries' experiences, we gain a broader understanding of the potential impact of administrative approval reforms on the technological complexity of manufacturing exports, particularly in promoting technological innovation, increasing export complexity, and enhancing international competitiveness.

We believe these additions further enrich the analys.

5. Improved Data Visualization: Future revisions should consider enhancing figures, charts, and tables used to present findings, ensuring they are accessible and informative. Clearer legends, color-coding, and detailed annotations could help convey complex data more effectively.

5.Thank you very much for your valuable suggestions. As our empirical analysis was conducted using Stata, all the figures were generated by the software, and unfortunately, we are unable to enhance the clarity of the data visualizations in this study. We will keep your recommendations in mind for future research and ensure the use of software capable of producing higher-quality visualizations.

---

## [Decision Letter · Decision Letter 3]

7 Feb 2025

An analysis of the impact of administrative approval reform on the technological complexity of manufacturing exports

PONE-D-24-27061R3

Dear Dr. Zhang,

We’re pleased to inform you that your manuscript has been judged scientifically suitable for publication and will be formally accepted for publication once it meets all outstanding technical requirements.

Kind regards,

Tinggui Chen

Academic Editor

PLOS ONE

Additional Editor Comments (optional):

Reviewers' comments:

Reviewer's Responses to Questions

**Comments to the Author**

1. If the authors have adequately addressed your comments raised in a previous round of review and you feel that this manuscript is now acceptable for publication, you may indicate that here to bypass the “Comments to the Author” section, enter your conflict of interest statement in the “Confidential to Editor” section, and submit your "Accept" recommendation.

Reviewer #1: (No Response)

Reviewer #2: All comments have been addressed

Reviewer #3: All comments have been addressed

2. Is the manuscript technically sound, and do the data support the conclusions?

Reviewer #1: (No Response)

Reviewer #2: Yes

Reviewer #3: Yes

3. Has the statistical analysis been performed appropriately and rigorously? 

Reviewer #1: (No Response)

Reviewer #2: Yes

Reviewer #3: Yes

4. Have the authors made all data underlying the findings in their manuscript fully available?

Reviewer #1: (No Response)

Reviewer #2: Yes

Reviewer #3: Yes

5. Is the manuscript presented in an intelligible fashion and written in standard English?

Reviewer #1: (No Response)

Reviewer #2: Yes

Reviewer #3: Yes

6. Review Comments to the Author

Reviewer #1: I have checked the revised mnauscript and corresponding response letter. The quality of this paper has been imporved. It can be accepted now.

Reviewer #2: The paper makes contribution to the field, and the writing is clear and well-organized. I find no major concerns and recommend acceptance of the paper in its current form.

Reviewer #3: After a careful check of the manuscript, I confirm a significant revision by the authors to the above issues based on the following:

Theoretical Framework: The paper extends the work of Hallak and Sivadasan (2013) by presenting administrative approval reform as an instrument for reducing institutional transaction costs. This innovative perspective contributes significantly to the theoretical understanding of export technological complexity.

Empirical Evidence: The authors utilize a robust econometric model and rich datasets to support their findings. The use of sensitivity tests and alternative metrics enhances the credibility of their results.

Policy Implications: The study offers concrete policy recommendations that address how to further optimize administrative processes to encourage technological advancement and improve export quality.

Depth of Analysis: The manuscript includes a thorough examination of the heterogeneous impacts of reforms across different types of enterprises and geographical regions, making it a nuanced contribution to the literature.

Methodological Rigor: The application of robust techniques such as the Heckman two-step procedure for addressing sample selection bias and Difference-in-Differences (DID) approach adds reliability to the findings.

7. PLOS authors have the option to publish the peer review history of their article (what does this mean?). If published, this will include your full peer review and any attached files.

Reviewer #1: No

Reviewer #2: No

Reviewer #3: **Yes: **Martial Agbor Fanga

---

## [Editor Report · Acceptance letter]

PONE-D-24-27061R3

PLOS ONE

Dear Dr. Zhang,

I'm pleased to inform you that your manuscript has been deemed suitable for publication in PLOS ONE. Congratulations! Your manuscript is now being handed over to our production team.

Kind regards,

on behalf of

Dr. Tinggui Chen

Academic Editor

PLOS ONE